# Representation Ensembling for Synergistic Lifelong Learning with Quasilinear Complexity

## Abstract

In lifelong learning, data are used to improve performance not only on the current task, but also on previously encountered, and as yet unencountered tasks. While typical transfer learning algorithms can improve performance on future tasks, their performance on prior tasks degrades upon learning new tasks (called forgetting). Many recent approaches for continual or lifelong learning have attempted to *maintain* performance on old tasks given new tasks. But striving to avoid forgetting sets the goal unnecessarily low. The goal of lifelong learning should be to use data to improve performance on both future tasks (forward transfer) and past tasks (backward transfer). Our key insight is that we can ensemble representations that were learned independently on disparate tasks to enable both forward and backward transfer, with algorithms that run in quasilinear time. Our algorithms demonstrate both forward and backward transfer in a variety of simulated and benchmark data scenarios, including tabular, vision (CIFAR-100, 5-dataset, Split Mini-Imagenet, and Food1k), and audition (spoken digit), including adversarial tasks, in contrast to various reference algorithms, which typically failed to transfer either forward or backward, or both.

## 1 Introduction

Learning is a process by which an intelligent system improves performance on a given task by leveraging data (Mitchell, 1999). In classical machine learning, the system is often optimized for a single task (Vapnik & Chervonenkis, 1971; Valiant, 1984). While it is relatively easy to *simultaneously* optimize for multiple tasks (multi-task learning) (Caruana, 1997), it has proven much more difficult to *sequentially* optimize for multiple tasks (Thrun, 1996; Thrun & Pratt, 2012). Specifically, classical machine learning systems, and natural extensions thereof, exhibit "catastrophic forgetting" when trained sequentially, meaning their performance on the prior tasks drops precipitously upon training on new tasks (McCloskey & Cohen, 1989; McClelland et al., 1995). However, learning could be lifelong, with agents continually building on past knowledge and experiences, improving on many tasks given data associated with any task. For example, in humans, learning a second language often improves performance in an individual's native language (Zhao et al., 2016).

In the past 30 years, a number of sequential task learning algorithms have attempted to overcome catastrophic forgetting. These approaches naturally fall into one of two camps. In one, the algorithm has fixed resources, and so must reallocate resources (essentially compressing representations) in order to incorporate new knowledge (Kirkpatrick et al., 2017; Zenke et al., 2017; Li & Hoiem, 2017; Schwarz et al., 2018; Finn et al., 2019). For efficient compression of the representation, the model weights can be regularized by using information extracted from old tasks or by directly replaying the old task data. Biologically, this corresponds to adulthood, where brains have a nearly fixed or decreasing number of cells and synapses. In the other, the algorithm adds (or builds) resources as new data arrive (essentially ensembling representations) (Ruvolo & Eaton, 2013; Rusu et al., 2016; Lee et al., 2019). Biologically, this corresponds to development, where brains grow by adding cells, synapses, etc. A close resemblance to this resource growing approach can be found in Sodhani et al. (2020), where the model adaptively expands when the capacity of the model saturates.

Approaches from both camps demonstrate some degree of continual (or lifelong) learning (Parisi et al., 2019). In particular, they can sometimes learn new tasks while not catastrophically forgetting old tasks (see Appendix A for a detailed discussion on the relevant algorithms). However, as we will show, many reference lifelong learning algorithms are unable to transfer knowledge forward (to future unseen tasks) and most of them do not transfer backward (to previously seen tasks). With high enough sample sizes, some of them are able to transfer forward or backward, but transfer is more important in low sample size regimes (Chen & Liu, 2016; Lee et al., 2019). This inability to effectively transfer in low-sample size regimes has been identified as one of the key obstacles limiting the capabilities of artificial intelligence (Pearl, 2019; Marcus & Davis, 2019). We focus primarily on the (arguably simpler) resource growing camp in which each new task is learned with additional representational capacity.

In this paper, We propose two lifelong learning algorithms, one based on ensembling decision forests (Synergistic Forests, SynF), and another based on ensembling deep networks (Synergistic Networks, SynN). We explore our proposed algorithm as compared to a number of reference algorithms on an extensive suite of numerical experiments that span simulations, vision datasets including CIFAR-100, 5-dataset, Split Mini-Imagenet, and Food1k, as well as the spoken digit dataset. Figure 1 illustrates that our algorithms outperform all the reference algorithms in terms of forward, backward, and overall transfer in most of the cases. Ablation studies indicate the degree to which the amount of representation or storage capacity and replaying old task data impact performance of our algorithms. All our code and experiments are open source to facilitate reproducibility.

## 2 Mathematical Framework

### 2.1 The lifelong learning objective

Consider a lifelong learning environment with tasks, $\mathcal{T} = \{1, 2, \cdots, T\}$. We consider task aware lifelong learning, i.e., the tasks are known during both training and testing time. For simplicity, we consider that any task $t \in \mathcal{T}$ has the same input space, i.e., they have $\mathcal{X} \subset \mathbb{R}^D$ valued inputs with $\mathcal{Y} = \{1, \cdots, K_t\}$ valued class labels. We assume the tasks arrive sequentially, but the data samples, $\mathbf{s}^t = \{(x_i, y_i)\}_{i=1}^{n_t}$ within each task $t$ are batched and sampled identically and independently ($iid$) from some fixed distribution. Here each sample within $\mathbf{s}^t$ is the realization of random variable pair, $(X, Y) \overset{iid}{\sim} \mathcal{D}_t$ and $\mathbf{s}^t$ is the realization of $\mathbf{S}^t$ distributed as the joint distribution $\mathcal{D}^n$. A learner $f \in \mathcal{F}$ trains on $\mathbf{s}^t$ and chooses a hypothesis $h \in \mathcal{H}$ by minimizing a particular risk, where $\mathcal{F}$ and $\mathcal{H}$ are the algorithm and hypothesis space, respectively. In supervised learning settings, one can consider the following risk for a particular task $t$:

$$R^t(f(\mathbf{S}^t)) = R^t(h_n) = \mathbb{E}_{(X,Y)\sim\mathcal{D}_t}[\ell_t(h(X), Y)], \tag{1}$$

where $\ell_t : \mathcal{Y} \times \mathcal{Y} \to [0, \infty)$ is a given loss function associated with the task $t$ and $h = f(\mathbf{S}^t)$. Note that the data $\mathbf{S}^t$ may contain data that is relevant to any number of tasks (potentially all the tasks) in the environment. One may take expectation with respect to $\mathcal{D}^n$ for averaging out the randomness in the risk due to $\mathbf{S}^t$ and consider the generalization error for the task as:

$$\mathcal{E}_f^t(\mathbf{S}^t) = \mathbb{E}_{\mathbf{S}^t\sim\mathcal{D}^n}[R^t(f(\mathbf{S}^t))]. \tag{2}$$

In the above equation, the learner will have access to a total of $T$ datasets after $T$ tasks, $\bigcup_{t=1}^T \mathbf{S}^t$, instead of $\mathbf{S}^t$ only. [1] The goal is to find a learner $f \in \mathcal{F}$ that chooses a hypothesis $h$ such the generalization error over all the tasks after observing all the data is minimized, that is:

$$\begin{array}{ll} \text{minimize} & \sum_{t=1}^T \mathcal{E}_f^t(\bigcup_{t'=1}^T \mathbf{S}^{t'}) \\ \text{subject to} & f \in \mathcal{F} \end{array}. \tag{3}$$

Note that $h$ can operate under any task $t$, i.e., $h = \bigcup_{t=1}^T h_t$.

---

[1]More generally, we may have $J$ datasets, where $J \neq T$ and each dataset may be associated with the target distributions of multiple tasks. For simplicity, we do not consider such scenarios further at this time.

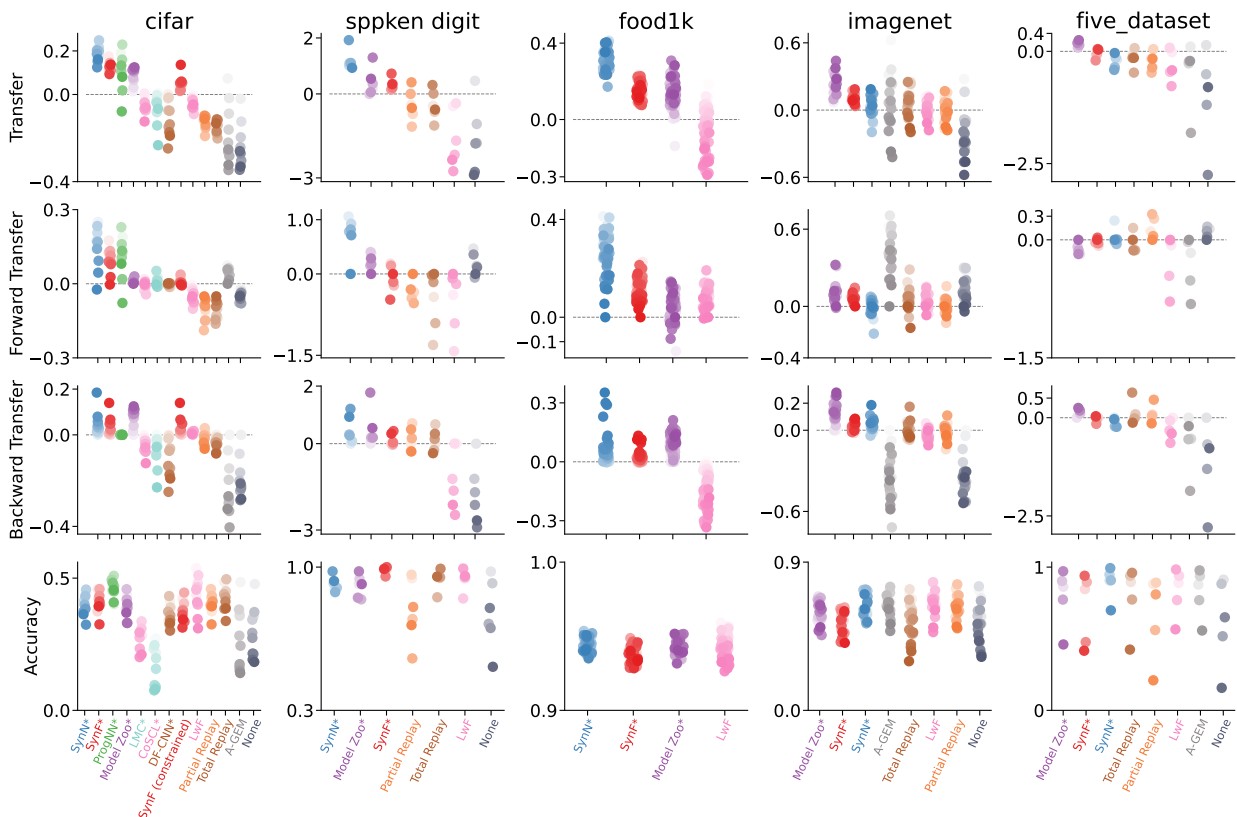

Figure 1: **Performance summary on vision and audition benchmark datasets.** Columns are datasets, rows are different evaluation criteria (see Section 2 for definitions, and Section 6 for experimental details), each strip of colored dots corresponds to an algorithm (we introduce SynN and SynF here). Colors of dots fade with increasing task number. Datasets on the right have higher number of training samples per task than those in the left. Resource growing algorithms have a '*'. EWC, O-EWC, SI, TAG and ER always perform worse than LwF, and hence we do not show them in the summary plot. SynN and SynF tend to exhibit the most transfer across datasets.

## 2.2 Lifelong learning evaluation criteria

Others have previously introduced criteria to evaluate transfer, including forward and backward transfer (Lopez-Paz & Ranzato, 2017; Benavides-Prado et al., 2018; Díaz-Rodríguez et al., 2018; Veniat et al., 2020). Pearl Judea (2018) introduced the transfer benefit ratio, which builds directly off relative efficiency from classical statistics (Bickel & Doksum, 2015). We define three notions of transfer building on relative efficiency.

**Definition 1 (Transfer)** *Overall transfer of algorithm f for a given Task t is:*

$$\mathsf{Transfer}^t(f) := \log \frac{\mathcal{E}_f^t(\mathbf{S}^t)}{\mathcal{E}_f^t(\bigcup_{t'=1}^T \mathbf{S}^{t'})}. \tag{4}$$

*We say that an algorithm f has transferred to task t from all the tasks up to T if and only if* $\mathsf{Transfer}^t(f) > 0$.

Forward transfer quantifies how much performance a learner transfers forward to future tasks, given prior tasks.

**Definition 2 (Forward Transfer)** *The forward transfer of f for task t is :*

$$\mathsf{Forward\ Transfer}^t(f) := \log \frac{\mathcal{E}_f^t(\mathbf{S}^t)}{\mathcal{E}_f^t(\bigcup_{t'=1}^t \mathbf{S}^{t'})}. \tag{5}$$

*We say an algorithm (positively) forward transfers for task t if and only if* $\mathsf{Forward\ Transfer}^t(f) > 0$.

Backwards transfer quantifies how much a learner transfers backward to previously observed tasks, in light of new tasks.

**Definition 3 (Backward Transfer)** *The backward transfer of f for Task t is:*

$$\mathsf{Backward\ Transfer}^t(f) := \log \frac{\mathcal{E}_f^t(\bigcup_{t'=1}^t \mathbf{S}^{t'})}{\mathcal{E}_f^t(\bigcup_{t'=1}^T \mathbf{S}^{t'})}. \tag{6}$$

*We say an algorithm (positively) backward transfers to Task t from all the tasks T if and only if* $\mathsf{Backward\ Transfer}^t(f) > 0$.

Note that $\mathsf{Transfer}$ can be decomposed into $\mathsf{Forward\ Transfer}$ and $\mathsf{Backward\ Transfer}$:

$$\mathsf{Transfer}^t(f) = \log \frac{\mathcal{E}_f^t(\mathbf{S}^t)}{\mathcal{E}_f^t(\bigcup_{t'=1}^T \mathbf{S}^{t'})} = \log \frac{\mathcal{E}_f^t(\mathbf{S}^t)}{\mathcal{E}_f^t(\bigcup_{t'=1}^t \mathbf{S}^{t'})} + \log \frac{\mathcal{E}_f^t(\bigcup_{t'=1}^t \mathbf{S}^{t'})}{\mathcal{E}_f^t(\bigcup_{t'=1}^T \mathbf{S}^{t'})} \tag{7}$$

$$= \mathsf{Forward\ Transfer}^t(f) + \mathsf{Backward\ Transfer}^t(f). \tag{8}$$

Another paper, Veniat et al. (2020), concomitantly introduced transfer and forgetting (backward transfer). Their statistics are the same as ours, except they do not use a log. We opted for a log to address numerical stability issues in comparing small numbers. Because log is a monotonic function, the order of ranking algorithms is preserved (Appendix Figure 1 shows a version of Figure 1 but using Veniat's statistics, which is nearly visually identical). By virtue of introducing $\mathsf{Forward\ Transfer}$ here, we can identify the inherent trade-off between forward and backward transfer, for a fixed amount of total transfer. Apart from the above statistics, we also report accuracy per task.

**Definition 4 (Accuracy)** *The accuracy of f on task t is after observing total T datasets is:*

$$\mathsf{Accuracy}^t(f) := 1 - \mathcal{E}_f^t(\bigcup_{t'=1}^T \mathbf{S}^{t'}). \tag{9}$$

## 3    Representation ensembling algorithms

Shannon proposed that a learner can be decomposed into three components: an encoder, a channel, and a decoder (Cover & Thomas, 2012; Cho et al., 2014): $h(\cdot) = w \circ v \circ u(\cdot)$. Figure 2 shows these three components as the building blocks of different learning schemas. The encoder, $u : \mathcal{X} \mapsto \tilde{\mathcal{X}}$, maps an $\mathcal{X}$-valued input into an internal representation space $\tilde{\mathcal{X}}$ (Vaswani et al., 2017; Devlin et al., 2018). The channel $v : \tilde{\mathcal{X}} \mapsto \Delta_{\mathcal{Y}}$ maps the transformed data into a posterior distribution (or, more generally, a score). Finally, a decoder $w : \Delta_{\mathcal{Y}} \mapsto \mathcal{Y}$, produces a predicted label.

A canonical example of a single learner depicted in Figure 2A is a decision tree. Importantly, one can subsample the training data to learn different components of the tree (Breiman et al., 1984; Denil et al., 2014; Athey et al., 2019). For example, one can use a portion of data to learn the tree structure (which is the encoder). Then, by pushing the remaining data (sometimes called the 'out-of-bag' data) through the tree, one can learn posteriors in each leaf node (which are the channel). The channel thus gives scores for each data point denoting the probability of that data point belonging to a specific class. Using separate sets of data to learn the encoder and the channel results in less bias in the estimated posterior in the channels as

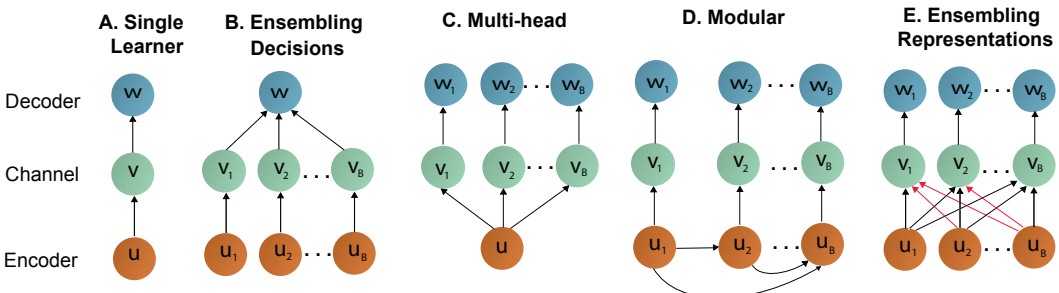

Figure 2: Schemas of composable hypotheses. A. Single task learner. B. Ensembling decisions (as output by the channels) is a well-established practice, including random forests and gradient boosted trees. C. Learning a joint representation or D. Ensembling representations (learned by the encoders) was previously used in lifelong learning scenarios, but were not trained independently as in E, thereby causing interference or forgetting. Note that the new encoders interact with the previous encoders through the channel layer (indicated by red arrows), thereby, enabling backward transfer. Again the old encoders interact with the future encoders (indicated by black arrows), thereby, enabling forward transfer.

in 'honest trees' (Breiman et al., 1984; Denil et al., 2014; Athey et al., 2019). Finally, the decoder provides the predicted class label using $\arg\max$ over the posteriors from the channel.

One can generalize the above decomposition by allowing for multiple encoders, as shown in Figure 2B. Given $B$ different encoders, one can attach a single channel to each encoder, yielding $B$ different channels. Doing so requires generalizing the definition of a decoder so that it would operate on multiple channels. Such a decoder ensembles the *decisions*, because here each channel provides the final output based on the encoder. This is the learning paradigm behind bagging (Breiman, 1996) and boosting (Freund, 1995); indeed, decision forests are a canonical example of a decision function operating on an ensemble of $B$ outputs (Breiman, 2001).

Although the task specific structure in Figure 2B can provide useful decision on the corresponding task, they cannot, in general, provide meaningful decisions on other tasks, because those tasks might have completely different class labels. Therefore, in the multi-head structure (Figure 2C) a single encoder is used to learn a joint representation from all the tasks, and a separate channel is learned for each task to get the score or class conditional posteriors for each task, which is followed by each task specific decider (Kirkpatrick et al., 2017; Schwarz et al., 2018; Zenke et al., 2017).

Modular approaches, such as PROGNN and LMC (Figure 2D), have both multiple encoders and decoders. Connections from past to future encoders enables forward transfer. However, they freeze backward transfer.

Our approach also uses multiple encoders and decoders (Figure 2E). Unlike modular approaches, we allow connections across encoders to other channels, including both forwards and backwards. The result is that the channels **ensemble representations** (learned by the encoders), rather than decisions (learned by the channels). In our algorithms, we push all the data through each encoder, and each channel learns and ensembles across all encoders. When each encoder has learned complementary representations, the channels can leverage that information to improve over single task performance. This approach has applications, particularly in few-shot (Dvornik et al., 2020) and multiple task scenarios, including lifelong learning.

### 3.1 Our representation ensembling algorithms

We have developed two different representation ensembling algorithms based on bagging which are trained on different tasks. We draw analogy from the synergy of multiple independent representations on a particular task and call our algorithm 'synergistic'. In both algorithms, as data from a new task arrives, the algorithm first builds a new independent encoder. Then, it builds the channel for this new task by pushing the new task data through *all* existing encoders. Thus the channel integrates information across all existing encoders using the new task data, thereby enabling forward transfer. At the same time, if it stores old task data (or can generate such data), it can push that data through the new encoders to update the channels from the

old tasks, thereby enabling backward transfer. In either case, new test data are passed through all existing encoders and corresponding channels to make a prediction (see Appendix C for detailed description of this approach).

### 3.1.1 Synergistic Networks

A Synergistic Network (SynN) ensembles deep networks. For each task, the encoder $u_t$ in SynN is the "backbone" of a deep network (DN). Thus, each $u_t$ maps an element of $\mathcal{X}$ to an element of $\mathbb{R}^d$, where $d$ is the number of neurons in the ultimate layer of the DN. The channels are learned via $k$-Nearest Neighbors ($k$-NN) (Stone, 1977) over the $d$ dimensional representations of $\mathcal{X}$. Recall that a $k$-NN, with $k$ chosen such that as the number of samples goes to infinity, $k$ also goes to infinity, while $\frac{k}{n} \to 0$, is a universally consistent classifier (Stone, 1977). We use $k = 16 \log_2 n$, which satisfies these conditions. The decoder $w_t$ outputs the argmax to produce a single prediction.

### 3.1.2 Synergistic Forests

Synergistic Forests (SynF) ensemble decision trees or forests. For each task, the encoder $u_t$ of SynF is the representation learned by a decision forest (Amit & Geman, 1997; Breiman, 2001). The leaf nodes of each decision forest partition the input space $\mathcal{X}$ into polytopes (Breiman et al., 1984). The channel then learns the class-conditional posteriors by populating the polytopes with out-of-task samples, as in "honest trees" (Breiman et al., 1984; Denil et al., 2014; Athey et al., 2019). Each channel outputs the posteriors averaged across the collection of forests learned over different tasks. The decoder $w_t$ outputs the argmax to produce a single prediction.

Note that the amount of additional representation capacity added per task by SynF is a function of the amount and complexity of the data for a new task. Contrast this with SynN and other deep net based modular or representation ensembling approaches, which *a priori* choose how much additional representation to add, prior to seeing all the task data. So, SynF has capacity, space complexity, and time complexity scale with the complexity and sample size of each task. In contrast, ProgNN, SynN (and others like it) have a fixed capacity for each task, even if the tasks have very different sample sizes and complexities.

## 4 A computational taxonomy of lifelong learners

The space complexity of the learner refers to the amount of memory space needed to store the learner (Kuo & Zuo, 2003). We also study the representation capacity of these algorithms. Capacity is defined as the size of the subset of hypotheses that is achievable by the learning algorithm (Zhang et al., 2021).

We use the soft-O notation $\tilde{\mathcal{O}}$ to quantify complexity (van Rooij et al., 2019). Letting $n$ be the sample size and $T$ be the number of tasks, we write that the capacity, space or time complexity of a lifelong learning algorithm is $f(n, t) = \tilde{\mathcal{O}}(g(n, T))$ when $|f|$ is bounded above asymptotically by a function $g$ of $n$ and $T$ up to a constant factor and polylogarithmic terms. For simplifying the calculation, we make the following assumptions:

1. Each task has the same number of training samples.

2. Capacity grows linearly with the number of trainable parameters in the model.

3. The number of epochs is fixed for each task, independent of sample size.

4. For the algorithms with dynamically expanding capacity, we assume the worst case scenario where an equal amount of capacity is added to the hypothesis with an additional task.

Assumption 3 enables us to write time complexity as a function of the sample size. Table 1 summarizes the capacity, space and time complexity of several reference algorithms, as well as our SynN and SynF. For space and time complexity, the table shows results as a function of $n$ and $T$, as well as the common scenario

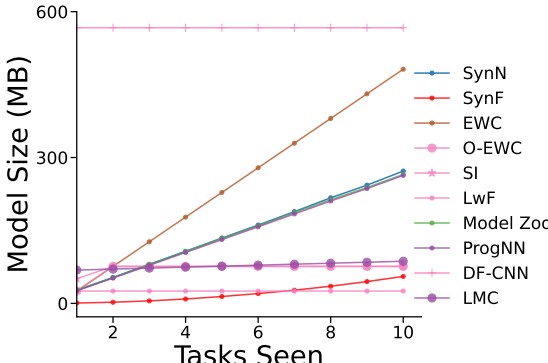

Figure 3: **Storage space as a function on number of tasks in CIFAR 10X10.** Memory consumed by `SynN` is dominated by the encoder size. The size of `DF-CNN` remains constant throughout.

Table 1: Capacity, space, and time complexity of the representation learned by various lifelong learning algorithms. We show soft-O notation ($\tilde{\mathcal{O}}(\cdot, \cdot)$ defined in main text) as a function of $n = \sum_t^T n_t$ and $T$, as well as the common setting where $n$ is proportional to $T$. The bottom three rows show algorithms whose space and time both grow quasilinearly with capacity growing.

| Parametric | Capacity | Space | | Time | | Examples |
|---|---|---|---|---|---|---|
| | $(n, T)$ | $(n, T)$ | $(n \propto T)$ | $(n, T)$ | $(n \propto T)$ | |
| parametric | 1 | 1 | 1 | $n$ | $n$ | `O-EWC`, `SI`, `LwF` |
| parametric | 1 | $T$ | $n$ | $nT$ | $n^2$ | `EWC` |
| parametric | 1 | $n$ | $n$ | $nT$ | $n^2$ | `TOTAL REPLAY` |
| semi-parametric | $T$ | $T^2$ | $n^2$ | $nT$ | $n^2$ | `PROGNN` |
| semi-parametric | $T$ | $T$ | $n$ | $n$ | $n$ | `DF-CNN` |
| semi-parametric | $T$ | $T+n$ | $n$ | $n$ | $n$ | `SYNN`, `MODEL ZOO`, `DER`, `LMC` |
| non-parametric | $n$ | $n$ | $n$ | $n$ | $n$ | `SYNF`, `IBP-WF` |

where sample size per task is fixed and therefore proportional to the number of tasks, $n \propto T$. For detailed calculation of time complexity see Appendix E.

Parametric lifelong learning methods have a representational capacity which is invariant to sample size and task number. Although the space complexity of some of these algorithms grow (because the number of the constraints grows stored by the algorithms grows, or they continue to store more data), their capacity is fixed. Thus, given a sufficiently large number of tasks, in general, eventually all parametric methods will catastrophically forget. `EWC` (Kirkpatrick et al., 2017), `ONLINE EWC` (Schwarz et al., 2018), `SI` (Zenke et al., 2017), and `LwF` (Li & Hoiem, 2017) are all examples of parametric lifelong learning algorithms.

Semi-parametric algorithms' representational capacity grows slower than sample size. For example, if $T$ is increasing slower than $n$ (e.g., $T \propto \log n$), then algorithms whose capacity is proportional to $T$ are semi-parametric. `PROGNN` (Rusu et al., 2016) is semi-parametric, nonetheless, its space complexity $\tilde{\mathcal{O}}(T^2)$ due to the lateral connections. Moreover, the time complexity for `PROGNN` also scales quadratically with $n$ when $n \propto T$. Thus, an algorithm that literally stores all the data it has ever seen, and retrains a fixed size network on all those data with the arrival of each new task, would have smaller space complexity and the same time complexity as `PROGNN`. For comparison, we implement such an algorithm and refer to it as Total Replay. `DF-CNN` (Lee et al., 2019) improves upon `PROGNN` by introducing a "knowledge base" with lateral connections to each new column, thereby avoiding all pairwise connections. Because these semi-parametric methods have a fixed representational capacity per task, they will either lack the representation capacity to perform well given sufficiently complex tasks, and/or will waste resources for very simple tasks. `SYNN` and `SYNF` eliminate the lateral connections between columns of the network, thereby reducing space complexity down to $\tilde{\mathcal{O}}(T)$. Moreover, as shown in Figure 3, memory consumed by new channels is negligible compared

to that of memory required for storing the encoders. Note that the time required for updating channels is negligible in comparison with the time required for training a new encoder.

Non-parametric algorithms' representational capacity grows linearly with sample size. SynF is a non-parametric lifelong learning algorithm with its capacity, space and time complexity all $\tilde{\mathcal{O}}(n)$, meaning that its representational capacity naturally increases with the complexity of each task. Apart from SynF, Indian Buffet Process for Weight Factors (IBP-WF) (Mehta et al., 2021) proposed the only other non-parametric lifelong learning algorithm to our knowledge.

# 5 Providing intuition of synergistic learning through simulations

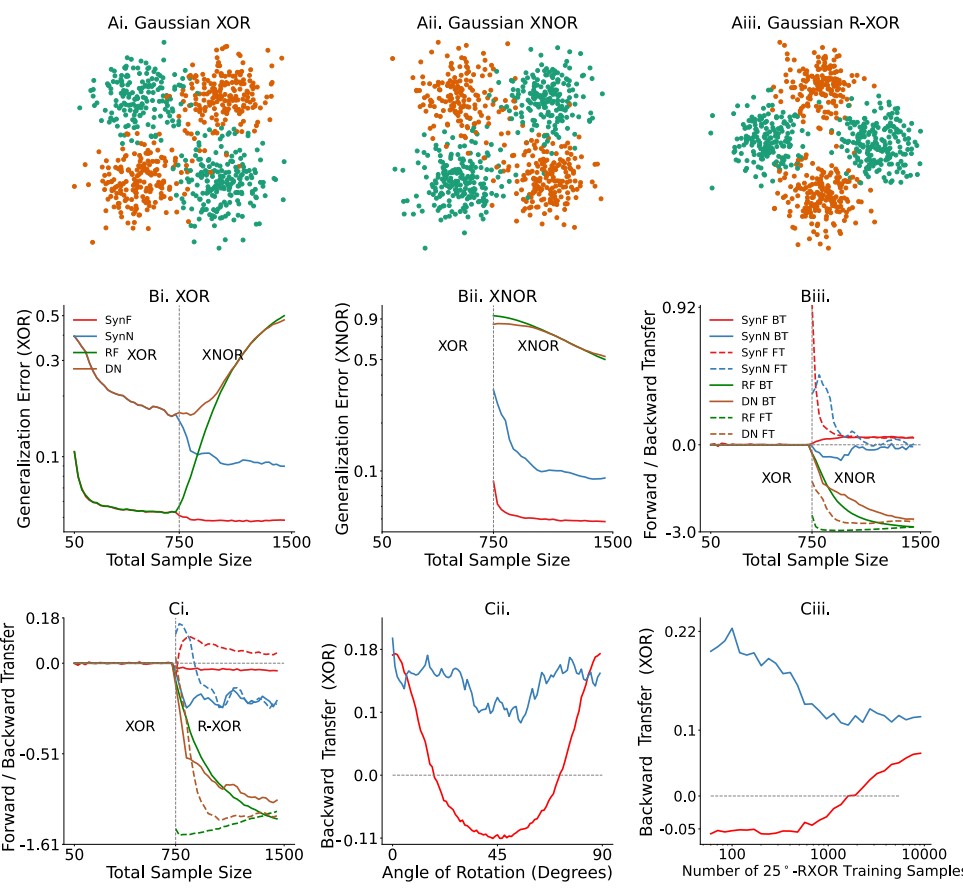

Figure 4: **Synergistic Forest and Synergistic Network demonstrate forward and backward transfer.** (*A*) 750 samples from: (*Ai*) Gaussian XOR, (*Aii*) XNOR, which has the same optimal discriminant boundary as XOR, and (*Aiii*) R-XOR, which has a discriminant boundary that is uninformative, and therefore adversarial, to XOR. (*Bi*) Generalization error for XOR, and (*Bii*) XNOR. SynF (SynN) outperforms RF (DN) on XOR when XNOR data is available, and on XNOR when XOR data are available. (*Biii*) Forward and backward transfer of SynF are positive for all sample sizes, and are negative for all sample sizes for RF. Forward and backward transfer for SynN is higher than DN for all sample sizes. (*Ci*) In an adversarial task setting, SynF and SynN gracefully forgets XOR, whereas RFand DN catastrophically forget and interfere. (*Cii*) Backward Transfer is positive with respect to XOR when the optimal decision boundary of $\theta$-XOR is similar to that of XOR (e.g. angles far from 45°), and negative otherwise.. (*Ciii*) Backward Transfer is a nonlinear function of the source training sample size (XOR sample size is fixed at 500).

### 5.1 Forward and backward transfer in a simple environment

Consider a very simple two-task environment: Gaussian XOR and Gaussian Exclusive NOR (XNOR) (Figure 4A, see Appendix F for details). The two tasks share the exact same discriminant boundaries: the coordinate axes. Thus, transferring from one task to the other merely requires learning a bit flip of the class labels. We sample a total 750 samples from XOR, followed by another 750 samples from XNOR.

SynF and random forests (RF) achieve the same generalization error on XOR when training with XOR data (Figure 4Bi). But because RF does not account for a change in task, when XNOR data appear, RF performance on XOR deteriorates (it catastrophically forgets). In contrast, SynF continues to improve on XOR given XNOR data, demonstrating backward transfer. Now consider the generalization error on *XNOR* (Figure 4Bii). Both SynF and RF are at chance levels for XNOR when only XOR data are available. When XNOR data are available, RF must unlearn everything it learned from the XOR data, and thus its performance on XNOR starts out nearly maximally inaccurate, and quickly improves. On the other hand, because SynF can leverage the encoder learned using the XOR data, upon getting *any* XNOR data, it immediately performs quite well, and then continues to improve with further XNOR data, demonstrating forward transfer (Figure 4Biii). SynF demonstrates positive forward and backward transfer for all sample sizes, whereas RF fails to demonstrate forward or backward transfer, and eventually catastrophically forgets the previous tasks. Results for SynN and DN are qualitatively similar to those of SynF and RF respectively.

### 5.2 Forward and backward transfer for adversarial tasks

In the context of synergistic learning, we informally define a task $t$ to be adversarial with respect to task $t'$ if the true joint distribution of task $t$, without any domain adaptation, impedes performance on task $t'$. In other words, training data from task $t$ can only add noise, rather than signal, for task $t'$. An adversarial task for Gaussian XOR is Gaussian XOR rotated by 45° (R-XOR) (Figure 4Aiii). Training on R-XOR therefore impedes the performance of SynF and SynN on XOR, and thus backward transfer becomes negative, demonstrating graceful forgetting (Aljundi et al., 2018) (Figure 4Ci). Because R-XOR is more difficult than XOR for SynF (because the discriminant boundaries are oblique (Tomita et al., 2020)), and because the discriminant boundaries are learned imperfectly with finite data, data from XOR can actually improve performance on R-XOR, and thus forward transfer is positive. In contrast, both forward and backward transfer are negative for RF and DN.

To further investigate this relationship, we design a suite of R-XOR examples, generalizing R-XOR from only 45° to any rotation angle between 0° and 90°, sampling 100 points from XOR, and another 100 from each R-XOR (Figure 4Cii). As the angle increases from 0° to 45°, Backward Transfer flips from positive ($\approx 0.18$) to negative ($\approx -0.11$) for SynF. A similar trend is also visible for SynN. The 45°-XOR is the maximally adversarial R-XOR. Thus, as the angle further increases, Backward Transfer increases back up to $\approx 0.18$ at 90°, which has an identical discriminant boundary to XOR. Moreover, when $\theta$ is fixed at 25°, Backward Transfer increases at different rates for different sample sizes of the source task (Figure 4Ciii).

Together, these experiments indicate that the amount of transfer can be a complicated function of (i) the difficulty of learning good representations for each task, (ii) the relationship between the two tasks, and (iii) the sample size of each. Appendix F further investigates this phenomenon in a multi-spiral environment.

## 6 Benchmark data experiments

For benchmark data, we build SynN encoders using the network architecture described in van de Ven et al. (2020). We use the same network architecture for all the benchmarking models. For the following experiments, we consider two modalities of real data: vision and language.

### 6.1 Reference algorithms

We compared our approaches to 15 reference lifelong learning methods. Among them five are resource growing as well as modular approach: ProgNN (Rusu et al., 2016), DF-CNN (Lee et al., 2019), LMC (Ostapenko et al., 2021), Model Zoo(Ramesh & Chaudhari, 2021), CoSCL (Wang et al., 2022). Other reference algorithms

are resource constrained: Elastic Weight Consolidation (`EWC`) (Kirkpatrick et al., 2017), Online-EWC (`O-EWC`) (Schwarz et al., 2018), Synaptic Intelligence (`SI`) (Zenke et al., 2017), Learning without Forgetting (`LwF`) (Li & Hoiem, 2017), and "None".

We also compare two variants of exact replay (Total Replay and Partial Replay) using the code provided by van de Ven et al. (2020). Both Total and Partial Replay store all the data they have ever seen, but Total Replay replays all of it upon acquiring a new task, whereas Partial Replay replays $M$ samples, randomly sampled from the entire corpus, whenever we acquire a new task with $M$ samples. Additionally, we have compared our approach with more constrained ways of replaying old task data, including Averaged Gradient Episodic Memory (`A-GEM`) (Chaudhry et al., 2018), Experience Replay (`ER`) (Chaudhry et al., 2019) and Task-based Accumulated Gradients (`TAG`) (Malviya et al., 2021).

For the baseline "None", the network was incrementally trained on all tasks in the standard way while always only using the data from the current task. The implementations for all of the algorithms are adapted from open source codes (Lee et al., 2019; van de Ven & Tolias, 2019); for implementation details, see Appendix D.

## 6.2 Exploring and explaining transfer capabilities via the CIFAR 10x10 dataset

The CIFAR 100 challenge (Krizhevsky, 2012), consists of 50,000 training and 10,000 test samples, each a 32x32 RGB image of a common object, from one of 100 possible classes, such as apples and bicycles. CIFAR 10x10 divides these data into 10 tasks, each with 10 classes (Lee et al., 2019) (see Appendix G for details).

### 6.2.1 Resource growing experiments

`SynF`, `SynN`, and `Model Zoo` demonstrate positive forward and backward transfer for every task in CIFAR 10x10, in contrast, other algorithms do not exhibit any positive backward transfer (Figure 1 first column). Moreover, they retained their accuracy while improving transfer (Figure 1, bottom row). `ProgNN` had a similar degree of forward transfer, but zero backward transfer, and requires quadratic space and time in sample size, unlike `SynF`, `SynN`, and `Model Zoo` which all require quasilinear space and time.

### 6.2.2 Ablation Experiments

Our proposed algorithms can improve performance on all the tasks (past and future) by both growing additional resources and replaying data from the past tasks. Below we do three ablation experiments using CIFAR 10X10 to measure the relative contribution of resource growth and replay on the performance of our proposed algorithms.

**Resource constrained experiments**   In this experiment, we devised a "resource constrained" variant of `SynF` experiments to observe the effect of ablating resource growth on `SynF`. In this constrained variant, we compare the lifelong learning algorithm to its single task variant, but ensure that they both have the same amount of resources. For example, on Task 2, we would compare `SynF` with 20 trees (10 trained on 500 samples from Task 1, and another 10 trained on 500 samples from Task 2) to `RF` with 20 trees (all trained on 500 samples Task 2). Although ablating the resource growth results in lower forward transfer compared to that of its resource growing variant (Figure 1 first column and Figure 5 top left), forward transfer remains positive after enough tasks, and backward transfer is actually invariant to this change (Figure 5, top left and center). In contrast, all of the reference algorithms that have fixed resources exhibit negative forward and backward transfer. Note that in this experiment, building the single task learners actually requires substantially *more* resources, specifically, $10 + 20 + \cdots + 100 = 550$ trees, as compared with only 100 trees in the prior experiments. In general, to ensure single task learners use the same amount of resources per task as our synergistic learners requires $O(n^2)$ resources, where as `SynF` only requires $O(n)$.

**Resource Recycling Experiments**   The binary distinction we made above, algorithms either build resources or reallocate them, is a false dichotomy, and biologically unnatural. In biological learning, systems develop from building to fixed resources, as they grow from juveniles to adults. To explore this continuum of amount of resources to grow, we trained `SynF` on the first nine CIFAR 10x10 tasks using 50 trees per task, with 500 samples per task. For the tenth task, we could (i) select the 50 trees (out of the 450 existing trees)

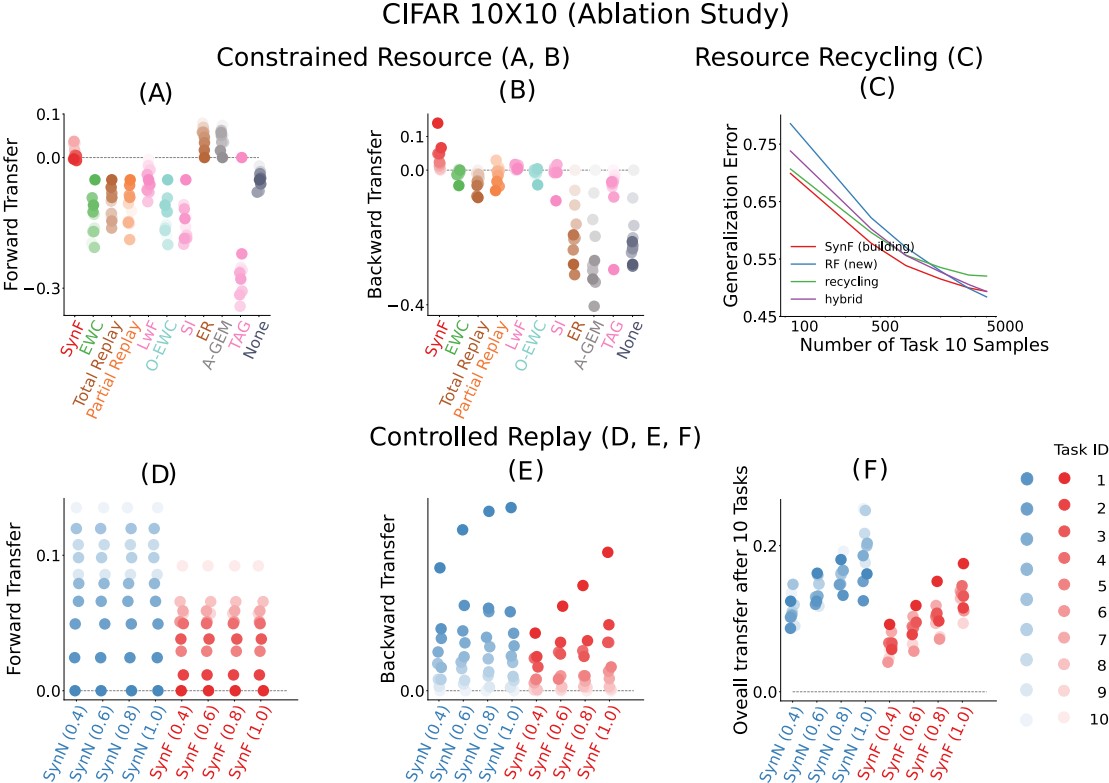

Figure 5: **Ablation experiments on SynN and SynF using CIFAR 10X10. (A, B)** Comparison of Resource Constrained SynF with algorithms having a fixed amount of resources. SynF is the only approach that demonstrate forward (*A*) and backward transfer (*B*). **(C)** Building and recycling ensembles are two boundaries of a continuum, with hybrid models in the middle. SynF achieves lower (better) generalization error than other approaches until 5,000 training samples on the new task are available, but eventually a hybrid approach wins. **(D, E, F)** Controlled replay experiment on CIFAR 10x10. Fraction of total samples per task replayed is mentioned in parenthesis in the middle and the right plot. *Left*: forward transfer is invariant to replay. *Middle*: Backward transfer increases as more samples are replayed from the old tasks. *Right*: Overall transfer increases with amount of replay.

that perform best on task 10 (recycling), (ii) train 50 new trees, as SynF would normally do (building), (iii) build 25 and recruit 25 trees (hybrid), or (iv) ignore all prior trees (RF). SynF outperforms other approaches except when 5,000 training samples are available, but the recycling approach is nearly as good as SynF (Figure 5, top right). This result motivates future work to investigate optimal strategies for determining how to optimally leverage existing resources without growing new ones given a new task.

**Controlled Replay Experiment** In this experiment, we train 4 different versions of SynN and SynF sequentially on the 10 tasks from CIFAR 10X10. The only difference between different versions of the algorithms is the amount of old task data replayed. In 4 different versions of each algorithm, we replay 40%, 60%, 80% and 100% of the old task data respectively. As apparent from Figure 5 bottom, replaying old task data has no effect on forward transfer, but replaying more data improves backward transfer as the number of tasks increases.

### 6.2.3 Adversarial experiments

Consider the same CIFAR 10x10 experiments above, but, for Tasks 2 through 9, randomly permute the class labels within each task, rendering each of those tasks adversarial with regard to the first task (because the

labels are uninformative). Figure 6A indicates that backward transfer for both SynF and SynN show positive backward transfer even with such label shuffling (the other algorithms did not demonstrate positive backward transfer). Now, consider a Rotated CIFAR experiment, which uses only data from the first task, divided into two equally sized subsets (making two tasks), where the second subset is rotated by different amounts (Figure 6, right). Backward transfer of both SynF and SynN is nearly invariant to rotation angle, whereas the other approaches are far more sensitive to rotation angle. Note that zero rotation angle corresponds to the two tasks *having identical distributions*. The fact that other algorithms fail to transfer even in this setting suggests that they may not ever be able to positively backwards transfer. See Appendix G.2 for additional experiment using CIFAR 10X10.

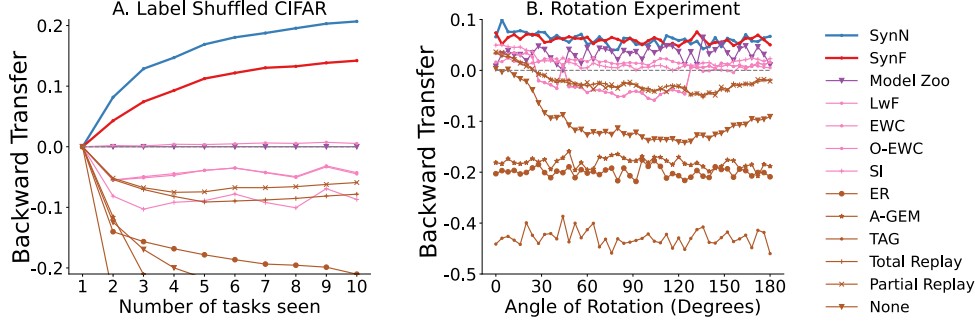

Figure 6: **Extended CIFAR 10x10 experiments.** *A*. Shuffling class labels within tasks two through nine with 500 samples each demonstrates both SynF and SynN can still achieve positive backward transfer, and that the other algorithms still fail to transfer. *B*. SynF and SynN are nearly invariant to rotations, whereas other approaches are more sensitive to rotation.

## 6.3 Further investigating transfer in additional datasets with more classes, tasks, and/or samples

### 6.3.1 Spoken Digit

In this experiment, we used the **Spoken Digit** dataset (Jackson et al., 2018). As shown in Figure 1 second column, both SynF and SynN show positive backward and forward transfer between the spoken digit tasks, in contrast to other methods, some of which show only forward transfer, others show only backward transfer, with none showing both, and some showing neither. See Appendix G.3 for details of the experiment.

### 6.3.2 FOOD1k 50X20 Dataset

In this experiment, we use **Food1k** which is a large scale vision dataset consisting of 1000 food categories from Food2k Min et al. (2021). FOOD1k 50X20 splits these data into 50 tasks with 20 classes each. For each class, we randomly sampled 60 samples per class for training the models and used rest of the data for testing purpose. Because on the CIFAR experiments Model Zoo performs the best among the reference resource growing models, and LwF is the best performing resource constrained algorithm, we only use them as the reference models for the large scale experiment to avoid heavy computational cost. As shown in Figure 1 third column, SynN performs the best among all the algorithms on this large dataset.

### 6.3.3 Split Mini-Imagenet

In this experiment, we have used the **Mini-Imagenet** dataset (Malviya et al., 2021). The dataset was split into 20 tasks with 5 classes each. Each task has 2400 training samples and 600 testing samples. As shown in Figure 1 fourth column, we get positive forward and backward transfer for both SynN and SynF. However, although samples per task is lower compared to that of 5-dataset, it is still quite high. Hence, Model Zoo outperforms all the algorithms in this experiment.

### 6.3.4   5-dataset

In this experiment, we have used **5-dataset** (Malviya et al., 2021). It consists of 5 tasks from five different datasets: CIFAR-10 (Krizhevsky, 2012), MNIST, SVHN (Netzer et al., 2011), notMNIST (Bulatov, 2011), Fashion-MNIST (Xiao et al., 2017). All the monochromatic images were converted to RGB format, and then resized to $3 \times 32 \times 32$. As shown in Appendix Table 4, training samples per task in 5-dataset is relatively higher than that of low data regime typically considered in lifelong learning setting. However, as shown in Figure 1 fifth column, SynN and SynF show less forgetting than most of the reference algorithms. On the other hand, Model Zoo shows comparatively better performance in relatively high task data size setup. Recall that SynN and SynF are based on bagging, and Model Zoo is based on boosting. It is well known that boosting often outperforms bagging when sample sizes are large [2]. In lifelong learning, we are often primarily concerned with situations in which we have a small number of samples per task.

## 7   Discussion

We introduced quasilinear representation ensembling as an approach to synergistic lifelong learning. Two specific algorithms, SynF and SynN, achieve both forward and backward transfer, by leveraging resources (encoders) learned for other tasks without undue computational burdens. In this paper, we have mainly focused on task-aware setting, because it is simpler. Future work will extend our approach to more challenging task-unaware settings. Ablation experiments with CIFAR 10x10 shows that Forest-based representation ensembling approaches can easily add new resources when appropriate. This work therefore motivates additional research on deep learning to enable dynamically adding resources when appropriate, and resuse the older representations like the moduler methods (Yoon et al., 2017; Mallya & Lazebnik, 2018; Veniat et al., 2020; Ostapenko et al., 2021).

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

# A Literature review

Prior work illustrates that ensembling learners can yield huge advantages in a wide range of applications. For example, in classical machine learning, ensembling trees leads to state-of-the-art random forest (Breiman, 2001) and gradient boosting tree algorithms (Chen & Guestrin, 2016). Similarly, ensembling networks shows promising results in various real-world applications (Qiu et al., 2014; Potes et al., 2016). Wang et al. (2003) used weighted ensemble of learners in a streaming setting with distribution shift. TrAdaBoost (Dai et al., 2007) boosts ensemble of learners to enable transfer learning. In continual learning scenarios, many algorithms have been built on these ideas by ensembling dependent representations. For example, Learn++ (Polikar et al., 2001) boosts ensembles of weak learners learned over different data sequences in class incremental lifelong learning settings (van de Ven et al., 2022). Model Zoo (Ramesh & Chaudhari, 2021) uses the same boosting approach in task incremental lifelong learning scenarios.

Another group of algorithms, ProgNN (Rusu et al., 2016) and DF-CNN (Lee et al., 2019) learn a new "column" of nodes and edges with each new task, and ensembles the columns for inference (such approaches are commonly called 'modular' now). The primary difference between ProgNN and DF-CNN is that ProgNN has forward connections to the current column from all the past columns. This creates the possibility of forward transfer while freezing backward transfer. However, the forward connections in ProgNN render it computationally inefficient for a large number of tasks. DF-CNN gets around this problem by learning a common knowledge base and thereby, creating the possibility of backward transfer.

Recently, many other modular approaches have been proposed in the literature that improve on ProgNN's capacity growth. These methods consider the capacity for each task being composed of modules that can be shared across tasks and grown as necessary. For example, PackNet Mallya & Lazebnik (2018) starts with a fixed capacity network and trains for additional tasks by freeing up portion of the network capacity using iterative pruning. Veniat et al. (2020) trains additional modules with each new task, and the old modules are only used selectively. Ostapenko et al. (2021) improved the memory efficiency of the modular methods by adding new modules according to the complexity of the new tasks. Mehta et al. (2021) proposed non-parametric factorization of the layer weights that promotes sharing of the weights between tasks. However, all of modular methods described above lack backward transfer because the old modules are not updated with the new tasks. Dynamically Expandable Representation (DER) (Yan et al., 2021) proposed an improvement over the modular approaches where the model capacity is dynamically expanded and the model is fine-tuned by replaying a portion of the old task data along with the new task data. This approach achieves backward transfer between tasks as reported by the authors in the experiments.

Another strategy for building lifelong learning machines is to use total or partial replay (van de Ven et al., 2020; Robins, 1995; Shin et al., 2017). Replay approaches keep the old data and replay them when faced with new tasks to mitigate catastrophic forgetting. However, as we will illustrate, previously proposed replay algorithms do not demonstrate positive backward transfer in our experiments, though they often do not forget as much as other approaches.

Our approach builds directly on previously proposed modular and replay approaches with one key distinction: in our approach, representations are learned independently. Empirically, for low sample sizes random forests (which learn independent trees) typically outperform gradient boosted trees (which learn dependent trees) (Caruana et al., 2004; 2008; Fernández-Delgado et al., 2014). Because our approach of representation ensembling is similar to that of random forest, we expect learning independent representations to outperform learning dependent representations in these scenarios as well. This phenomenon is empirically shown in the main text Figure 1. Independent representations also have computational advantages, as doing so merely requires quasilinear time and space, and can be learned in parallel.

# B  Evaluation Criteria

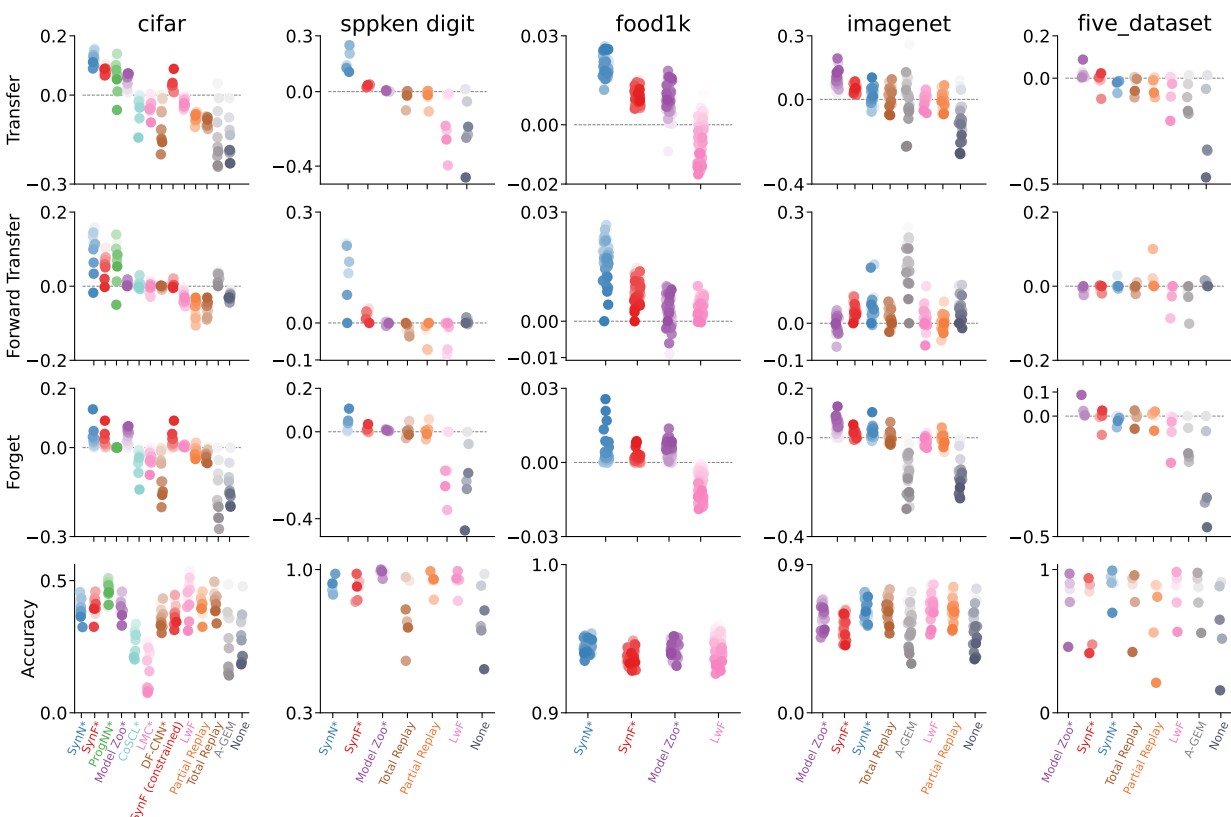

Figure 1: **Performance summary on vision and audition benchmark datasets using Veniat et al. (2020)'s statistics.** See Figure 1 for caption details. Note that the results here look nearly identical other than the y-axis labels.

# C  Representation Ensembling Algorithms

In this paper, we have proposed two representation ensembling algorithms, Synergistic Forests (SynF) and Synergistic Networks (SynN). The two algorithms differ in their details of how to update encoders and channels, but abstracting a level up they are both special cases of the same procedure. Let SynX refer to any possible synergistic algorithm. Algorithms 1, 2, 3, and 4 provide pseudocode for adding encoders, updating channels, and making predictions for any SynX algorithm. Whenever the learner gets access to a new task data, we use Algorithm 1 to train a new encoder for the corresponding task. We split the data into two portions — one set is used to learn the encoder and the other portion is called the held out or out-of-bag (OOB) data which is returned by Algorithm 1 to be used by Algorithm 2 to learn the channel for the corresponding task. Note that we push the OOB data through the in-task encoder and the whole dataset through the cross-task encoders to update the channel, i.e, learn the posteriors according to the new encoder. Then we use Algorithm 3 to replay the old task data through the new encoder and update their corresponding channels. Finally, while predicting for a test sample, we use Algorithm 4. Given the task identity, we use the corresponding channel to get the average estimated posterior and predict the class label as the arg max of the estimated posteriors.

---

**Algorithm 1** Add a new SynX encoder for a task. OOB = out-of-bag.

---

**Require:**
    (1) $t$                                                        ▷ current task number
    (2) $\mathcal{D}_n^t = (\mathbf{x}^t, \mathbf{y}^t) \in \mathbb{R}^{n \times p} \times \{1, \ldots, K\}^n$          ▷ training data for task $t$
**Ensure:**
    (1) $u_t$                                      ▷ an encoder trained on task $t$
    (2) $\mathcal{I}_{OOB}^t$                                ▷ a set of the indices of OOB data
 1: **function** SynX.FIT($t, (\mathbf{x}^t, \mathbf{y}^t)$)
 2:     $u_t, \mathcal{I}_{OOB}^t \leftarrow$ encoder.fit($\mathbf{x}^t, \mathbf{y}^t$)          ▷ train an encoder on partitioned data
 3:     **return** $u_t, \mathcal{I}_{OOB}^t$
 4: **end function**

---

**Algorithm 2** Add a new SynX channel for the current task.

---

**Require:**
    (1) $t$                                                       ▷ current task number
    (2) $\boldsymbol{u}_t = \{u_t\}_{t'=1}^t$                              ▷ the set of encoders
    (3) $\mathcal{D}_n^t = (\mathbf{x}^t, \mathbf{y}^t) \in \mathbb{R}^{n \times p} \times \{1, \ldots, K\}^n$         ▷ training data for task $t$
    (4) $\mathcal{I}_{OOB}^t$               ▷ a set of the indices of OOB data for the current task
**Ensure:** $\boldsymbol{v}_t = \{v_{t,t'}\}_{t'=1}^t$        ▷ in-task ($t' = t$) and cross-task ($t' \neq t$) channels for task $t$
 1: **function** SynX.ADD__CHANNEL($t, \boldsymbol{u}_t, (\mathbf{x}_t, \mathbf{y}_t), \mathcal{I}_{OOB}^t$)
 2:     $v_{tt} \leftarrow u_t$.add__channel($(\mathbf{x}_t, \mathbf{y}_t), \mathcal{I}_{OOB}^t$)      ▷ add the in-task channel using OOB data
 3:     **for** $t' = 1, \ldots, t-1$ **do**           ▷ update the cross task channels for task $t$
 4:         $v_{tt'} \leftarrow u_{t'}$.add__channel($\mathbf{x}_t, \mathbf{y}_t$)
 5:     **end for**
 6:     **return** $v_t$
 7: **end function**

---

**Algorithm 3** Update SynX channel for the previous tasks.

---

**Require:**
    (1) $t$                                                     ▷ current task number
    (2) $u_t$                                   ▷ encoder for the current task
    (3) $\mathcal{D} = \{\mathcal{D}^{t'}\}_{t'=1}^{t-1}$            ▷ training data for tasks $t' = 1, \cdots, t-1$
**Ensure:** $\boldsymbol{v} = \{\boldsymbol{v}_{t'}\}_{t'=1}^{t-1}$                   ▷ all previous task voters
 1: **function** SynX.UPDATE__CHANNEL($t, u_t, \mathcal{D}$)
 2:     **for** $t' = 1, \ldots, t-1$ **do**           ▷ update the cross task channels
 3:         $v_{t't} \leftarrow u_t$.get__channel($\mathbf{x}_{t'}, \mathbf{y}_{t'}$)
 4:     **end for**
 5:     **return** $\boldsymbol{v}$
 6: **end function**

---

---

**Algorithm 4** Predicting a class label using SYNX.

**Require:**
    (1) $x \in \mathbb{R}^p$                                                       ▷ test datum
    (2) $t$                              ▷ task identity associated with $x$
    (3) $\boldsymbol{u}$                                ▷ all $T$ reperesenters
    (4) $\boldsymbol{v}_t$                              ▷ channel for task $t$
**Ensure:** $\hat{y}$                            ▷ a predicted class label
  1: **function** $\hat{y} = \text{SYNX.PREDICT}(t, x, v_t)$
  2:     $T \leftarrow \text{SYNX.get\_task\_number()}$             ▷ get the total number of tasks
  3:     $\hat{\mathbf{p}}_t = \mathbf{0}$                   ▷ $\hat{\mathbf{p}}_t$ is a $K$-dimensional posterior vector
  4:     **for** $t' = 1, \ldots, T$ **do**        ▷ aggregate the posteriors calculated from $T$-th task channel
  5:         $\hat{\mathbf{p}}_t \leftarrow \hat{\mathbf{p}}_t + v_{tt'}.\text{predict\_proba}(u_{t'}(x))$
  6:     **end for**
  7:     $\hat{\mathbf{p}}_t \leftarrow \hat{\mathbf{p}}_t / T$
  8:     $\hat{y} = \arg\max_i(\hat{\mathbf{p}}_t)$     ▷ find the index $i$ of the elements in the vector $\hat{\mathbf{p}}_t$ with maximum probability
  9:     **return** $\hat{y}$
10: **end function**

---

Table 1: Hyperparameters for SYNF in CIFAR-10X10 experiments. n_estimators is denoted by $B$, the number of trees, above.

| Hyperparameters | Value |
|---|---|
| n_estimators (500 training samples per task) | 10 |
| n_estimators (5000 training samples per task) | 40 |
| max_depth | 30 |
| max_samples (OOB split) | 0.67 |
| min_samples_leaf | 1 |

Table 2: Hyperparameters for SYNF in Five Datasets, Split Mini-Imagenet, FOOD1k experiments. n_estimators is denoted by $B$, the number of trees, above. Note that we use the same hyperparameters for all of the aforementioned datasets.

| Hyperparameters | Value |
|---|---|
| n_estimators | 10 |
| max_depth | 30 |
| max_samples (OOB split) | 0.67 |
| min_samples_leaf | 1 |

Table 3: Hyperparameters for SYNN in CIFAR 10X10, Five Datasets, Split Mini-Imagenet, FOOD1k experiments. Note that we use the same hyperparameters for all of the aforementioned datasets.

| Hyperparameters | Value |
|---|---|
| optimizer | Adam |
| learning rate | $3 \times 10^{-4}$ |
| max_samples (OOB split) | 0.67 |
| K (KNN channel) | $\log_2$(number of samples per task) |

## D  Reference Algorithm Implementation Details

The same network architecture was used for all compared deep learning methods. Following van de Ven et al. (2020), the 'base network architecture' consisted of five convolutional layers followed by two-fully connected layers each containing 2000 nodes with ReLU non-linearities and a softmax output layer. The convolutional layers had 16, 32, 64, 128 and 254 channels, they used batch-norm and a ReLU non-linearity, they had a 3x3 kernel, a padding of 1 and a stride of 2 (except the first layer, which had a stride of 1). This architecture was used with a multi-headed output layer (i.e., a different output layer for each task) for all algorithms using a fixed-size network. For ProgNN and DF-CNN the same architecture was used for each column introduced for each new task, and in our SynN this architecture was used for the transformers $u_t$ (see above). In these implementations, ProgNN and DF-CNN have the same architecture for each column introduced for each task. Among the reference algorithms, EWC, O-EWC, LwF, SI, Total Replay and Partial Replay results were produced using the repository `https://github.com/GMvandeVen/progressive-learning-pytorch`. For ProgNN and DF-CNN we used the code provided in `https://github.com/Lifelong-ML/DF-CNN`. For all other reference algorithms, we modified the code provided by the authors to match the deep net architecture as mentioned above and used the default hyperparameters provided in the code.

## E  Training Time Complexity Analysis

Consider a lifelong learning environment with $T$ tasks each with $n'$ samples, i.e., total training samples, $n = n'T$. For all the algorithm with time complexity $\tilde{\mathcal{O}}(n)$, the training time grows linearly with more training samples. We discuss all other algorithms with non-linear time complexity below.

### E.1  EWC

Consider the time required to train the weights for each task in EWC is $k_c n'$ and each task adds additional $k_l n'$ time from the regularization term. Here, $k_c$ and $k_l$ are both constants. Therefore, time required to learn all the $T$ tasks can be written as:

$$
\begin{aligned}
k_c n' &+ (k_c n' + k_l n') + \cdots + (k_c n' + (T-1)k_l n') \\
&= k_c n'T + k_l n' \sum_{t=1}^{T-1} t \\
&= k_c n'T + k_l n' \frac{T(T-1)}{2} \\
&= k_c n + 0.5 k_l nT - 0.5 k_l n \\
&= \tilde{\mathcal{O}}(nT).
\end{aligned}
\tag{10}
$$

### E.2  Total Replay

Consider the time to train the model on $n'$ samples is $k_c n'$. Therefore, time required to learn all the $T$ tasks can be written as:

$$
\begin{aligned}
k_c n' &+ k_c(n' + n') + \cdots + k_c n'T \\
&= k_c n' \sum_{t=1}^{T} t \\
&= k_c n' \frac{T(T+1)}{2} \\
&= 0.5 k_c nT + 0.5 k_c n \\
&= \tilde{\mathcal{O}}(nT)
\end{aligned}
\tag{11}
$$

### E.3 ProgNN

Consider the time required to train each column in ProgNN is $k_c n'$ and each lateral connection can be learned with time $k_l n'$. Therefore, time required to learn all the $T$ tasks can be written as:

$$
\begin{aligned}
k_c n' &+ (k_c n' + k_l n') + \cdots + (k_c n' + (T-1)k_l n') \\
&= k_c n' T + k_l n' \sum_{t=1}^{T-1} t \\
&= k_c n' T + k_l n' \frac{T(T-1)}{2} \\
&= k_c n + 0.5 k_l n T - 0.5 k_l n \\
&= \tilde{\mathcal{O}}(nT)
\end{aligned}
\tag{12}
$$

## F    Simulated Results

In each simulation, we constructed an environment with two tasks. For each, we sample 750 times from the first task, followed by 750 times from the second task. These 1,500 samples comprise the training data. We sample another 1,000 hold out samples to evaluate the algorithms. We fit a random forest (RF) (technically, an uncertainty forest which is an honest forest with a finite-sample correction (Mehta et al., 2019)) and a SynF. For SynN, we have used a deep network (DN) architecture with two hidden layers each having 10 nodes. Similarly, for SynN experiments we did 100 repetitions and reported the results after smoothing it using moving average with a window size of 5. For the SynF experiments we used 1000 repetitions and reported the mean of these repetitions. We repeat this process 30 times to obtain errorbars. Error bars in all cases were negligible.

### F.1   Gaussian XOR

Gaussian XOR is two class classification problem with equal class priors. Conditioned on being in class 0, a sample is drawn from a mixture of two Gaussians with means $\pm \begin{bmatrix} 0.5, & 0.5 \end{bmatrix}^\mathsf{T}$, and variances proportional to the identity matrix. Conditioned on being in class 1, a sample is drawn from a mixture of two Gaussians with means $\pm \begin{bmatrix} 0.5, & -0.5 \end{bmatrix}^\mathsf{T}$, and variances proportional to the identity matrix. Gaussian XNOR is the same distribution as Gaussian XOR with the class labels flipped. Rotated XOR (R-XOR) rotates XOR by $\theta°$ degrees.

### F.2   Spirals

A description of the distributions for the two tasks is as follows: let $K$ be the number of classes and $S \sim$ multinomial$(\frac{1}{K}\vec{1}_K, n)$. Conditioned on $S$, each feature vector is parameterized by two variables, the radius $r$ and an angle $\theta$. For each sample, $r$ is sampled uniformly in $[0, 1]$. Conditioned on a particular class, the angles are evenly spaced between $\frac{4\pi(k-1)t_K}{K}$ and $\frac{4\pi(k)t_K}{K}$ where $t_K$ controls the number of turns in the spiral. To inject noise along the spiral, we add Gaussian noise to the evenly spaced angles $\theta' : \theta = \theta' + \mathcal{N}(0, \sigma_K^2)$. The observed feature vector is then $(r\,\cos(\theta), r\,\sin(\theta))$. In Figure 2 we set $t_3 = 2.5$, $t_5 = 3.5$, $\sigma_3^2 = 3$ and $\sigma_5^2 = 1.876$.

Consider an environment with a three spiral and five spiral task (Figure 2). In this environment, axis-aligned splits are inefficient, because the optimal partitions are better approximated by irregular polytopes than by the orthotopes provided by axis-aligned splits. The three spiral data helps the five spiral performance because the optimal partitioning for these two tasks is relatively similar to one another, as indicated by positive forward transfer. This is despite the fact that the five spiral task requires more fine partitioning than the three spiral task. Because SynF grows relatively deep trees, it over-partitions space, thereby rendering tasks with more coarse optimal decision boundaries useful for tasks with more fine optimal decision boundaries. The five spiral data also improves the three spiral performance.

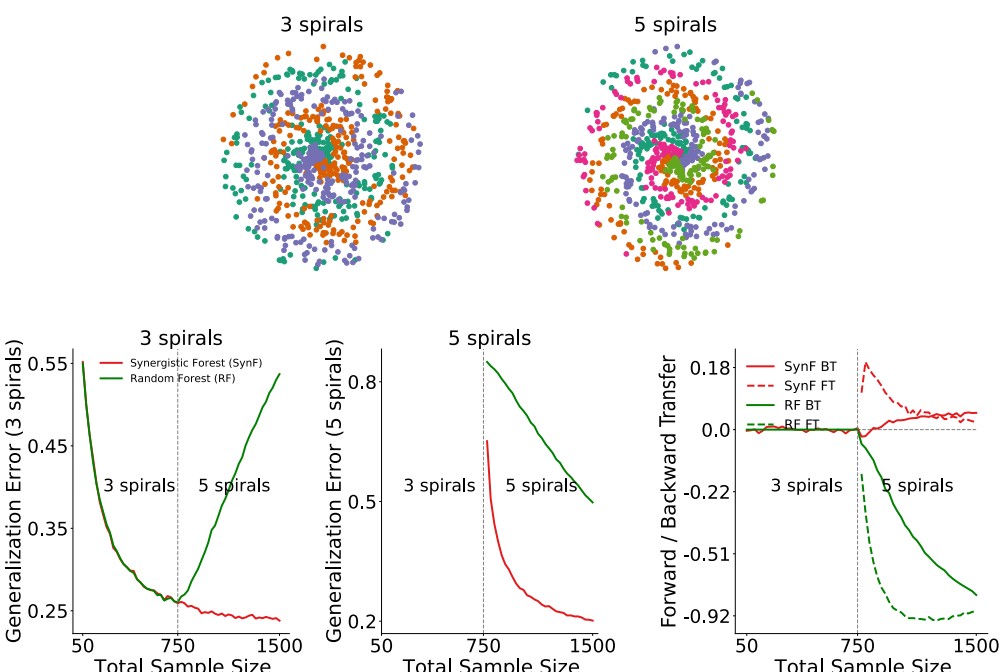

Figure 2: *Top*: 750 samples from 3 spirals (left) and 5 spirals (right). *Bottom left*: `SynF` outperforms `RF` on 3 spirals when 5 spirals data is available, demonstrating *backward* transfer in `SynF`. *Bottom center*: `SynF` outperforms `RF` on 5 spirals when 3 spirals data is available, demonstrating *forward* transfer in `SynF`. *Bottom right*: Transfer Efficiency of `SynF`. The forward (solid) and backward (dashed) curves are the ratio of the generalization error of `SynF` to `RF` in their respective figures. `SynF` demonstrates decreasing forward transfer and increasing backward transfer in this environment.

# G   Real Data Extended Results

**FOOD1k** and **Mini-Imagenet** datasets were obtained from `https://www.kaggle.com/datasets/whitemoon/miniimagenet` and `https://github.com/pranshu28/TAG`, respectively.

Table 4: Benchmark dataset details.

| Experiment | Dataset | Training samples | Testing samples | Dimension |
|---|---|---|---|---|
| CIFAR 10X10 | CIFAR 100 | 5000 | 10000 | $3 \times 32 \times 32$ |
| 5-dataset | CIFAR-10 | 50000 | 10000 | $3 \times 32 \times 32$ (resized) |
|  | MNIST | 60000 | 10000 |  |
|  | SVHN | 73257 | 26032 |  |
|  | notMNSIT | 16853 | 1873 |  |
|  | Fashion-MNIST | 60000 | 10000 |  |
| Split Mini-Imagenet | Mini-Imagenet | 48000 | 12000 | $3 \times 84 \times 84$ |
| FOOD1k 50X20 | Food1k | 60000 | 99682 | $3 \times 50 \times 50$ (resized) |
| Spoken Digit | Spoken Digit | 1650 | 1350 | $28 \times 28$ (processed and resized) |

## G.1   CIFAR 10x10

## G.2   CIFAR 10x10 Repeated Classes

We also considered the setting where each task is defined by a random sampling of 10 out of 100 classes with replacement. This environment is designed to demonstrate the effect of tasks with shared subtasks, which is a common property of real world lifelong learning tasks. Supplementary Figure 3 shows transfer of `SynF` and `SynN` on Task 1.

Table 5: **Performance metric: average** Transfer **after** 10 **tasks calculated for different algorithms on CIFAR 10x10** (500 **samples per task).**

| Algorithms | Transfer($\pm$std) |
| --- | --- |
| SynN | **0.19**($\pm$0.04) |
| SynF | **0.13**($\pm$0.02) |
| Model Zoo | 0.09($\pm$0.04) |
| ProgNN | 0.11($\pm$0.09) |
| LMC | $-$0.05($\pm$0.04) |
| CoSCL | $-$0.06($\pm$0.07) |
| DF-CNN | $-$0.11($\pm$0.08) |
| EWC | $-$0.15($\pm$0.04) |
| Total Replay | $-$0.15($\pm$0.03) |
| Partial Replay | $-$0.13($\pm$0.03) |
| SynF(resource constrained) | **0.05**($\pm$0.03) |
| LwF | $-$0.05($\pm$0.03) |
| O-EWC | $-$0.14($\pm$0.04) |
| SI | $-$0.16($\pm$0.03) |
| ER | $-$0.13($\pm$0.12) |
| A-GEM | $-$0.19($\pm$0.14) |
| TAG | $-$0.32($\pm$0.04) |
| None | $-$0.24($\pm$0.10) |

Table 6: **Performance metric: average** Transfer **after** 10 **tasks calculated for different algorithms on 5-dataset.**

| Algorithms | Transfer($\pm$std) |
| --- | --- |
| SynN | $-$**0.27**($\pm$0.22) |
| SynF | $-$**0.05**($\pm$0.11) |
| Model Zoo | 0.24($\pm$0.12) |
| EWC | $-$1.06($\pm$0.60) |
| Total Replay | $-$0.18($\pm$0.22) |
| Partial Replay | $-$0.27($\pm$0.26) |
| LwF | $-$0.39($\pm$0.48) |
| O-EWC | $-$1.07($\pm$0.60) |
| SI | $-$1.15($\pm$0.69) |
| ER | $-$0.78($\pm$1.03) |
| A-GEM | $-$0.55($\pm$0.90) |
| TAG | $-$0.56($\pm$0.58) |
| None | $-$1.15($\pm$1.30) |

Supplementary Table 10 shows the image classes associated with each task number.

### G.3  Spoken Digit experiment

In this experiment, we used the **Spoken Digit** dataset provided in `https://github.com/Jakobovski/free-spoken-digit-dataset`. The dataset contains audio recordings from six different speakers with 50 recordings for each digit per speaker (3000 recordings in total). The experiment was set up with six tasks where each task contains recordings from only one speaker. For each recording, a spectrogram was extracted using Hanning windows of duration 16 ms with an overlap of 4 ms between the adjacent windows. The spectrograms were resized down to $28 \times 28$. The extracted spectrograms from eight random recordings of '5' for six speakers are shown in Figure 5. For each Monte Carlo repetition of the experiment, spectrograms extracted for each task were randomly divided into 55% train and 45% test set. The experiment is summa-

Table 7: **Performance metric: average** Transfer **after** 10 **tasks calculated for different algorithms on Split Mini-Imagenet.**

| Algorithms | Transfer($\pm$std) |
|---|---|
| SYNN | **0.02**($\pm$0.10) |
| SYNF | **0.10**($\pm$0.04) |
| MODEL ZOO | 0.23($\pm$0.10) |
| EWC | $-0.29$($\pm$0.12) |
| TOTAL REPLAY | 0.06($\pm$0.13) |
| PARTIAL REPLAY | 0.00($\pm$0.10) |
| LwF | 0.02($\pm$0.08) |
| O-EWC | $-0.21$($\pm$0.10) |
| SI | $-0.14$($\pm$0.12) |
| ER | $-0.02$($\pm$0.27) |
| A-GEM | 0.06($\pm$0.26) |
| TAG | $-0.05$($\pm$0.15) |
| NONE | $-0.22$($\pm$0.23) |

Table 8: **Performance metric: average** Transfer **after** 10 **tasks calculated for different algorithms on FOOD1k 50X20.**

| Algorithms | Transfer($\pm$std) |
|---|---|
| SYNN | **0.31**($\pm$0.06) |
| SYNF | **0.14**($\pm$0.04) |
| MODEL ZOO | 0.13($\pm$0.08) |
| LwF | $-0.06$($\pm$0.13) |

rized in Figure 6. Note that we could not run the experiment on other 5 reference algorithms using the code provided by their authors.

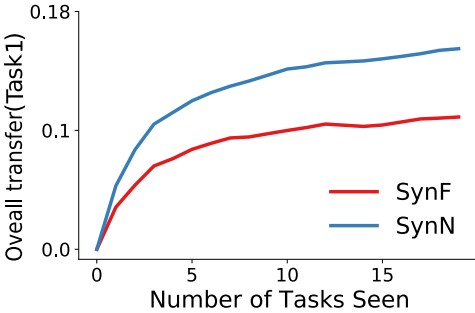

Figure 3: SYNF and SYNN transfer knowledge effectively when tasks share common classes. Each task is a random selection of 10 out of the 100 CIFAR-100 classes. Both SYNF and SYNN demonstrate monotonically increasing transfer efficiency for up to 20 tasks.

Table 9: Hyperparameters for SYNF in spoken digit experiment.

| Hyperparameters | Value |
|---|---|
| n_estimators (275 training samples per task) | 10 |
| max_depth | 30 |
| max_samples (OOB split) | 0.67 |
| min_samples_leaf | 1 |

Table 10: Task splits for CIFAR 10x10.

| Task # | Image Classes |
|---|---|
| 1 | apple, aquarium fish, baby, bear, beaver, bed, bee, beetle, bicycle, bottle |
| 2 | bowl, boy, bridge, bus, butterfly, camel, can, castle, caterpillar |
| 3 | chair, chimpanzee, clock, cloud, cockroach, couch, crab, crocodile, cup, dinosaur |
| 4 | dolphin, elephant, flatfish, forest, fox, girl, hamster, house, kangaroo, keyboard |
| 5 | lamp, lawn mower, leopard, lion, lizard, lobster, man, maple tree, motor cycle, mountain |
| 6 | mouse, mushroom, oak tree, orange, orchid, otter, palm tree, pear, pickup truck, pine tree |
| 7 | plain, plate, poppy, porcupine, possum, rabbit, raccoon, ray, road, rocket |
| 8 | rose, sea, seal, shark, shrew, skunk, skyscraper, snail, snke, spider |
| 9 | squirrel, streetcar, sunflower, sweet pepper, table, tank, telephone, television, tiger, tractor |
| 10 | train, trout, tulip, turtle, wardrobe, whale, willow tree, wolf, woman, worm |

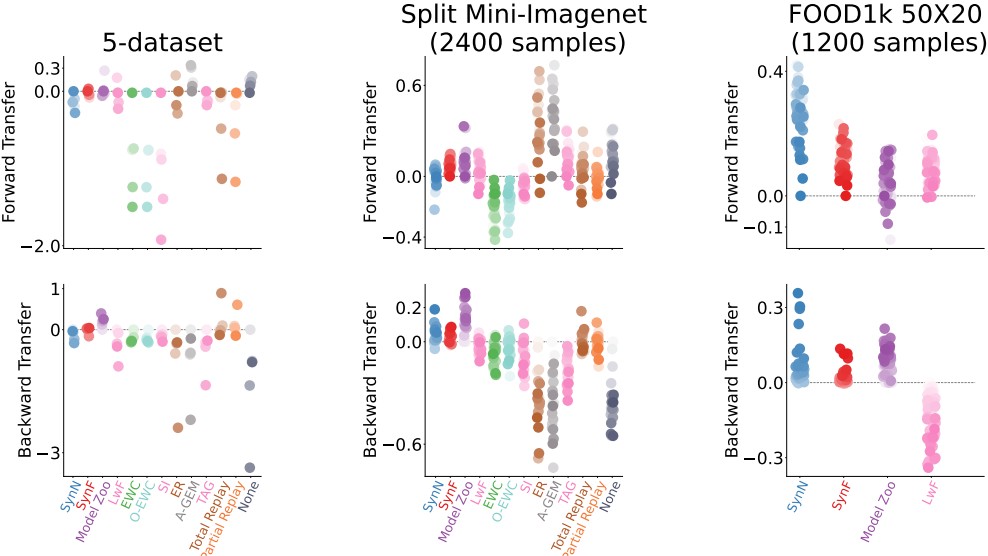

Figure 4: **Extended results on the different vision experiments.** This plot contains algorithms not shown in Figure 1.

Short-Time Fourier Transform Spectrogram of Number 5

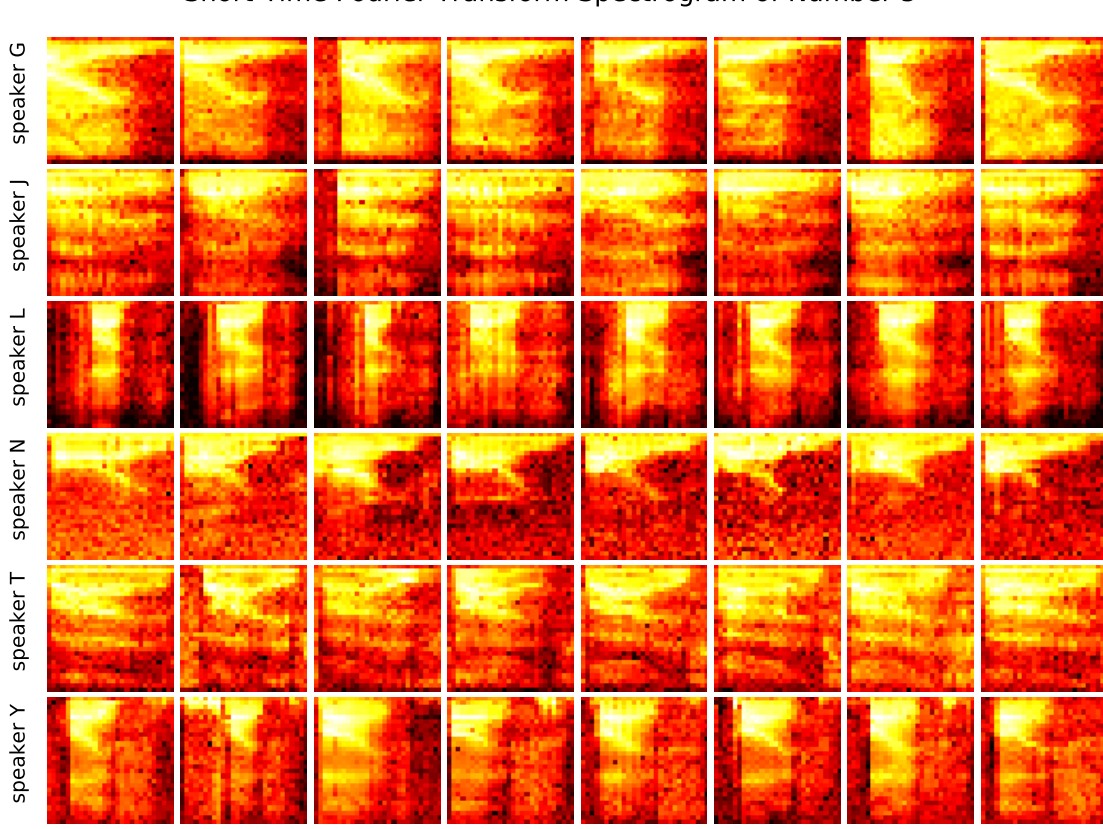

Figure 5: Spectrogram extracted from eight different recordings of six speakers uttering the digit 'five'.

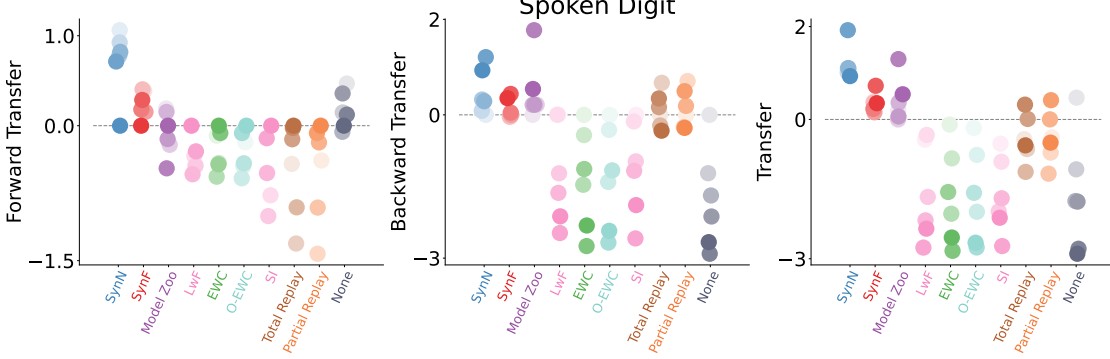

Figure 6: **Extended results on the Spoken Digit experiments.** This plot contains algorithms not shown in Figure 1.

