# OpenReview forum: "Representation Ensembling for Synergistic Lifelong Learning with Quasilinear Complexity"
_TMLR — Rejected by TMLR_

### Review · Reviewer_Xb3S · 2023-11-01

**Summary Of Contributions:**

This paper proposes a method for task-aware continual learning by training independent representation learners for each task and then ensemble the representations for prediction. I personally have not been convinced by the idea so can not summarise the contributions.

**Audience:**

Yes

**Claims And Evidence:**

Yes

**Requested Changes:**

I would suggest the authors rethink their ideas carefully.
The whole section 3 is unnecessary and the existing measurements are valid to evaluate different aspects of continual learning methods.

**Strengths And Weaknesses:**

As the method combines both modular and replay strategies, it highly relaxes the constraints of continual learning in different settings. For example, the existing methods of adding capacity to the model usually do not require storing samples from old tasks. With the assumption of knowing task identifiers,  simple multihead models can give reasonable performance without replay. The proposed method basically learns separate models for all tasks and ensemble their representations by KNN which requires replay. It is not cheap in model size and memory cost. The so-called backward/forward transferring only happens in the channel layer, which does not actually improve the learned representations.

The justification is not convincing. For example, the authors claim "Because transfer is particularly important in the low-sample size regime, we expect learning independent representations to outperform learning dependent representations in these scenarios as well." So why independent representations should outperform dependent representations in terms of knowledge transfer? In transfer learning and knowledge distillation, most methods share knowledge in representation learning.

The clarity of the evaluation criteria (Sec 3) is also poor. Section 3.1 seems unnecessarily long and it does not really relate to the proposed method. For example, I believe most researchers know that average accuracy can be skewed by one task, it does not need one paragraph to describe the issue. On the other hand, the description of the proposed performance measure is lack of clarity.  For example, what is the difference between $S$ and $S'$ in the definition 1? If it measures transfer efficiency, how can it be computed at a single time point? Shouldn't the transfer be estimated between a previous time point and a recent time point? The definition 2 does not make sense to me either. The denominator is the average error over all tasks and the numerator is the error of a single task, so how this metric can be larger than 1 for all tasks? There must be some tasks that have lower errors than the average error.  And how can it measure learning efficiency? What does "learning efficiency" exactly mean here? Efficiency in sample size, model size, or training time?  As the proposed measures are not clear enough to me, I cannot tell if the reported results are trustworthy or not.

---

> ### Author Response · Authors · 2024-01-13
> **Thanks for the feedback!**
>
> We appreciate the thoughtful feedback from reviewer Xb3S. We address your concerns below:
>  - "As the method combines both modular and replay strategies, it highly relaxes the constraints of .... simple multihead models can give reasonable performance without replay."
>
>  Reply: Yes, we have studied a simple setting, the same setting as in Model Zoo [2], CoSCL [3], and other recent lifelong learning papers. More challenging settings will be future work.
>
> - "The proposed method basically learns separate models for all tasks and ensemble their representations by KNN which requires replay. It is not cheap in model size and memory cost. "
>
> Reply: Thank you for your comment.  This was indeed unclear previously, so we added Figure 3, which shows that SynN grows at the same rate as other methods (Model Zoo and ProgNN), and smaller than EWC and DF-CNN.  And SynF starts out the smallest of all approaches, and after 10 tasks, is 2nd smallest. Moreover, as shown in the ablation study (Figure 5 top right), we can recycle the old resources and stop growing new encoders as well as updating channels after we have learned an adequate number of encoders. This shows the possibility of constant resource mode operation for our algorithms. However, we will pursue the constant resource operation in detail in future.
>
> - "The so-called backward/forward transferring only happens in ... So why independent representations should outperform dependent representations in terms of knowledge transfer? In transfer learning and knowledge distillation, most methods share knowledge in representation learning. "
>
> Reply: Thank you, this was unclear. We were referring to the empirical evidence in the literature which shows that bagging (independent representations) outperforms boosting (dependent representations) when sample size is small.  We removed that line from Section 1. Please see Section 6.3.4 and Appendix A last paragraph for clarification.
>
> - "The clarity of the evaluation criteria (Sec 3) is also poor. Section 3.1 seems unnecessarily long and it does not really relate to the proposed method. For example, I believe most researchers know that average accuracy can be skewed by one task, it does not need one paragraph to describe the issue. "
>
> Reply: Thank you, we significantly shortened this section, and removed those parts. The whole manuscript is now only 12.5 pages, down from 23 pages.
>
> - "On the other hand, the description of the proposed performance measure is lack of .... As the proposed measures are not clear enough to me, I cannot tell if the reported results are trustworthy or not."
>
> Reply: We regret we were not clear.  We have revised to clarify. Please see the updated Section 2 for clarification. Of note, our metrics are nearly identical to those of Veniat et al., 2020, though we added a logarithm for numerical stability (see Figure 1, Appendix Figure 1).
>
> - "The denominator is the average error over all tasks and the numerator is the error of a single task, so how this metric can be larger than 1 for all tasks?"
>
> Reply: The denominator is the error on a specific task $t$ when the learner has seen data from all the tasks.
>
>
> 1. Veniat, Tom, Ludovic Denoyer, and Marc'Aurelio Ranzato. "Efficient continual learning with modular networks and task-driven priors." arXiv preprint arXiv:2012.12631 (2020).
> 2. Ramesh, Rahul, and Pratik Chaudhari. "Model Zoo: A Growing" Brain" That Learns Continually." arXiv preprint arXiv:2106.03027 (2021).
> 3. Wang, Liyuan, et al. "Coscl: Cooperation of small continual learners is stronger than a big one." European Conference on Computer Vision. Cham: Springer Nature Switzerland, 2022.”

---

### Review · Reviewer_zumF · 2023-11-29

**Summary Of Contributions:**

The contributions mostly lie in two aspects: (1) a new series of metrics for evaluating the continual learning algorithms; (2) two new methods that can outperform existing algorithms in terms of the proposed metrics. The authors first overviewed the previous works of the continual learning community, and introduced the drawbacks of current evaluation pipelines. Based on that, they proposed two algorithms that can outperform the existing methods. Multiple datasets are used to benchmark the performance of the proposed methods.

**Audience:**

Yes

**Claims And Evidence:**

Yes

**Requested Changes:**

Proofreading; More explanation on why conducting the experiments; potentially compare with more recent algorithms.

**Strengths And Weaknesses:**

Strengths:
1. The motivation is clear;
2. The writing is mostly clear and coherent.
3. Intensive analysis was made by the authors for a comprehensive understanding

Weakness:

Some references have wrong formats. For example, on the second page 3rd paragraph the authors wrote "TrAdaBoost Dai et al. (2007)" -> should be "TrAdaBoost (Dai et al., 2007)". Please make sure the references have correct formats.

I think this method is similar to Mixture-of-Experts where the representations will be sent to different smaller modules and do feature mixture.

Regarding the experiments, I would suggest remake the 3D plots in Figure 6 and  Figure 9 which are hard to parse at the first glance. It would be better to have a numerical tables to present all these values, even though I understand it is to show the dynamics of backward transfer scores.

The section of single-task learner may need more explanation. What are the points the authors trying to show here?

There are more recent works, such as CoSCL [r1], which the authors fail to compare with. It would be helpful if the authors conduct experiments to compare with those more recent works.

[r1] CoSCL: Cooperation of Small Continual Learners is Stronger than a Big One

---

> ### Author Response · Authors · 2024-01-13
> **Thanks for the feedback!**
>
> We appreciate the thoughtful feedback from reviewer zumF. We address your concerns below:
>
> - "Some references have wrong formats..."
>
> Reply: We have fixed the formats.
>
> - "Regarding the experiments, I would suggest remake the 3D plots..."
>
> Reply: We remade the plots with the final transfer after all the tasks have been introduced (see Figure 1, 5, Appendix Figure 1, 4, 6).
>
> - "The section of single-task learner may need more explanation. What are the points the authors trying to show here?"
>
> Reply: After addressing other reviewers concerns, the single-task learner experiment seemed redundant, so we removed it.
>
> - "There are more recent works, such as CoSCL [r1], which the authors fail to compare with. It would be helpful if the authors conduct experiments to compare with those more recent works."
>
> Reply: We have added CoSCL as a baseline for the CIFAR-10X10 experiments (Figure 1). The conclusions remain unchanged.

---

### Review · Reviewer_eHsj · 2023-12-22

**Summary Of Contributions:**

This paper suggests ensembling representations from each task to address lifelong learning. By combining the encoders for each task, the proposed method is robust for both forward and backward transfers.

**Audience:**

Yes

**Claims And Evidence:**

Yes

**Requested Changes:**

- Rebuttal on the unresolved concerns from the previous submission.
- Rebuttal on my newly added comments.

**Strengths And Weaknesses:**

## Strength

- Addresses forward and backward transfers is an important issue in lifelong learning.
- Proposes new evaluation metrics to account for both forward and backward transfers.
- Presents various experimental results, spanning vision and speech domains.


## Weakness

My major concern is that the requested changes from the previous submission (https://openreview.net/forum?id=4ID37Uv64p) have not been fully resolved. The reviewers put substantial effort into providing detailed and constructive feedback. Before I introduce my additional concerns, I hope the authors will first provide a rebuttal regarding the unresolved concerns.
- There was no rebuttal for Reviewer K9wf's comments.
- While there was a reply to Reviewer szZq's comments, I feel the original concerns have not been fully addressed. The original concerns are as follows: 1) evaluation protocol is not convincing, 2)  baselines are far from SOTA, and 3) more direct comparisons with modularity-based methods should be included. The rebuttal partly addressed this issue by adding 1) a comparison with LMC and 2) a metric from Veniat et al. However, this is just a minimal rebuttal that touches the surface and does not address the fundamental concerns. For instance, I completely agree with the reviewer's claim that "I raise this point as: (1) there are already many existing metrics in CL literature which makes direct comparison more difficult and I believe we should go towards unifying rather than branching out, and (2) in the experimental section there are some experiments where the proposed method seems to be worse from the perspective of the "traditional" metrics but better from the perspective of the introduced metrics which makes it possible that metrics were chosen in order to fit the method well." The reviewer is explicitly requesting the comparison with SOTA modularity-based methods using "their" tables that report traditional metrics, which "go towards unifying rather than branching out."


Here are my additional concerns regarding the method and presentation:
- Training a new encoder for each task is not practical.
    - In a lifelong learning setup, an infinite number of tasks continuously arise. The proposed method requires training a new encoder for each task, resulting in quasilinear complexity, which is impractical. I acknowledge that this is also a challenge in previous works like ProgNN. However, recent works such as LMC have addressed this issue by reducing the computational cost. They achieve this by only updating a small module for each task, which is much smaller than the entire network. Consequently, the memory complexity becomes O(N + MT), where N represents the size of the shared backbone, M is the size of the module, and T is the number of tasks, with N >> M, making it manageable.
    - Thus, how to define the "tasks" is crucial. One may not need to train a separate encoder for similar tasks. Accessing the similarity and diversity of tasks to decide whether to create a new encoder or adapt existing ones would be a natural method design.
    - Another issue to consider is how to train the encoder. In these days, people commonly use a pre-trained backbone and fine-tune it for each task. Are the available data in the lifelong learning scenario sufficient to train the encoders from scratch? Integration with modern pre-trained models should be explored.
- Missing related works on representation ensembling for meta-learning.
    - It is a widely used practice to combine the backbones trained from known tasks to adapt them for the novel task [1]. This approach is more natural in a meta-learning scenario, where novel tasks are typically represented as interpolations of known tasks. However, it may be less suitable for a lifelong learning scenario since, by definition, the model should learn "new" concepts beyond what is already known. This is relevant to the definition of "tasks" that I mentioned above. Adding these discussions would be informative.
- Presentation is not polished.
    - As Reviewers K9wf and 8Tcb pointed out in the previous submission, "This paper has not been presented well, I had a difficult time navigating the paper to place and evaluate the algorithm in context." For example, the paragraphs in the introduction are organized in an encyclopedic manner rather than providing a crystallized flow to deliver the core message and storyline. This also applies to the overall paper structure. Please carefully assess whether the current section-subsection-paragraph structure is the best way to present the paper.
    - This paper exceeds 12 pages. While I acknowledge that many papers go beyond 8~12 pages to include all their content, I see the limited page count as beneficial. It encourages authors to focus on the most crucial information in the main text, reserving minor details for the appendix. It is the authors' responsibility to ensure concise content, rather than burdening readers with disorganized material.

[1] Selecting relevant features from a multi-domain representation for few-shot classification. ECCV'20.

---

> ### Author Response · Authors · 2024-01-13
> **Thanks for the feedback!**
>
> We appreciate the thoughtful feedback from reviewer eHsj. We address your concerns below:
>
> - "There was no rebuttal for Reviewer K9wf's comments."
>
> Reply: We are sorry that reviewer eHsj could not see the rebuttal for Reviewer K9wf’s comments. We have made the rebuttal visible to everyone to resolve the issue. However, as we will describe below, we have changed the draft significantly to address the concerns about the performance statistics.
>
> - "While there was a reply to Reviewer szZq's comments, I feel the original concerns... The reviewer is explicitly requesting the comparison with SOTA modularity-based methods using "their" tables that report traditional metrics, which "go towards unifying rather than branching out."
>
> Reply: Thank you, we have done that (see Appendix Figure 1). Upon further investigation, Veniat et al.’s forgetting and transfer are the same as our backwards and overall transfer, respectively, though we added a logarithm for numerical stability (see Section 2.2, Figure 1, Appendix Figure 1). Thus, their metrics just rescale ours (though we also added forward transfer, and illustrated that overall transfer decomposed into forward and backward transfer).  We also added CoSCL, a SOTA modularity-based method.  The conclusions remain unchanged.
>
> - "Training a new encoder for each task is not practical. In a lifelong learning setup, an infinite number of tasks continuously arise. The proposed method requires training..."
>
> Reply: This was indeed unclear previously, so we added Figure 3, which shows that SynN grows at the same rate as other methods (Model Zoo and ProgNN), and smaller than EWC and DF-CNN.  And SynF starts out the smallest of all approaches, and after 10 tasks, is 2nd smallest. Moreover, as shown in the ablation study (Figure 5 top right), we can recycle the old resources and stop growing new encoders as well as updating channels after we have learned an adequate number of encoders. This shows the possibility of constant resource mode operation for our algorithms. However, we will pursue the constant resource operation in detail in future.
>
> - "Thus, how to define the "tasks" is crucial. One may not need to train a separate encoder ..."
>
> Reply: Yes, that is future work.  For now, we operate in the same setting as Model Zoo, CoSCL, and other recent papers, where ‘task’ is provided to the learner.
>
> - "Another issue to consider is how to train the encoder. In these days, people commonly use a pre-trained backbone and fine-tune it for each task. Are the available data..."
>
> Reply: To our knowledge, the available data in lifelong learning scenarios are not sufficient to train the encoders from scratch.
>
> - "Missing related works on representation ensembling for meta-learning."
>
> Reply: We have cited the suggested work in the last paragraph of Section 3.
>
> - "Presentation is not polished..."
>
> Reply: Thank you, we have reorganized and shortened the paper to 12.5 pages from 23 pages, and reserved minor details for the appendix (which we also shortened).

---

### Decision · Action_Editor_GJsA · 2024-02-18

**Recommendation:** Reject

**Comment:**

This paper proposes two representation ensembling algorithms (SynN, SynF) using independent feature encoders for each task and all connections between channels and encoders for lifelong learning. The proposed algorithm empirically validates its better forward and backward transfers over other methods on several benchmark datasets.

The reviewers’ ratings are divergent. I thoroughly read the paper, reviews, and the authors’ responses. Overall, many concerns and claims are resolved and addressed, and the paper quality including the organization and the description is also improved. However, the following main concerns are still remained.
- Scalability of the algorithm
  - SynN, which seems to be a more practical algorithm considering the recent deep learning era, still has a limitation in scalability due to the independently trained encoder for each task from scratch with data from all tasks.
  - The performance of recycling in Fig. 5(c) is worse than that of the original SynF.
  - There is no strategy in recycling or stopping the expansion of encoders.
- Benefit in using independent encoders
  - There is no empirical analysis on the benefits in using independent encoders, especially from the perspective of performances, in this paper, not from reference papers indirectly. For example, what if we use dependent encoders while maintaining all connections between encoders and channels?
  - Intuitively, leveraging or transferring from previous encoders would be helpful especially in low-data cases. The authors did not fully address this point.
- Necessity of replay data
  - The proposed algorithm seems to be highly rely on the replay data from past tasks. However, the ablation experiments on this is insufficient. More detailed explanation on the results in Fig. 5(d,e,f) with more reduced replay ratio is necessary.
- Clear algorithm description
  - More clear descriptions on SynN, SynF are necessary. Especially, Algorithm 2 and 3 in Appendix C need to be improved. What are “add_channel” and “get_channel”?

Based on this, I think that the current submission does not meet the acceptance criteria of TMLR. I recommend the authors to resubmit the paper after a revision based on these comments.

**Audience:**

Some TMLR's audiences could be interested in this paper, since continual or lifelong learning is one of the important topics in current machine learning. However, the task-aware setting with the use of replay data in the paper could reduce their interest.

**Claims And Evidence:**

The following concerns are still remained.
- Scalability of the algorithm
  - SynN, which seems to be a more practical algorithm considering the recent deep learning era, still has a limitation in scalability due to the independently trained encoder for each task from scratch with data from all tasks.
  - The performance of recycling in Fig. 5(c) is worse than that of the original SynF.
  - There is no strategy in recycling or stopping the expansion of encoders.
- Benefit in using independent encoders
  - There is no empirical analysis on the benefits in using independent encoders, especially from the perspective of performances, in this paper, not from reference papers indirectly. For example, what if we use dependent encoders while maintaining all connections between encoders and channels?
  - Intuitively, leveraging or transferring from previous encoders would be helpful especially in low-data cases. The authors did not fully address this point.
- Necessity of replay data
  - The proposed algorithm seems to be highly rely on the replay data from past tasks. However, the ablation experiments on this is insufficient. More detailed explanation on the results in Fig. 5(d,e,f) with more reduced replay ratio is necessary.
- Clear algorithm description
  - More clear descriptions on SynN, SynF are necessary. Especially, Algorithm 2 and 3 in Appendix C need to be improved. What are “add_channel” and “get_channel”?

**Resubmission Of Major Revision:**

The authors may consider submitting a major revision at a later time.